# Polymer hetero-electrolyte enabled solid-state 2.4-V Zn/Li hybrid batteries

Ze Chen[1,6], Tairan Wang[1,6], Zhuoxi Wu[1], Yue Hou[1], Ao Chen[1], Yanbo Wang[1], Zhaodong Huang[1], Oliver G. Schmidt [2,3], Minshen Zhu [2,3] ✉, Jun Fan [1] ✉ & Chunyi Zhi [1,4,5] ✉

The high redox potential of $Zn^{0/2+}$ leads to low voltage of Zn batteries and therefore low energy density, plaguing deployment of Zn batteries in many energy-demanding applications. Though employing high-voltage cathode like spinel $LiNi_{0.5}Mn_{1.5}O_4$ can increase the voltages of Zn batteries, $Zn^{2+}$ ions will be immobilized in $LiNi_{0.5}Mn_{1.5}O_4$ once intercalated, resulting in irreversibility. Here, we design a polymer hetero-electrolyte consisting of an anode layer with $Zn^{2+}$ ions as charge carriers and a cathode layer that blocks the $Zn^{2+}$ ion shuttle, which allows separated Zn and Li reversibility. As such, the Zn∥LNMO cell exhibits up to 2.4 V discharge voltage and 450 stable cycles with high reversible capacity, which are also attained in a scale-up pouch cell. The pouch cell shows a low self-discharge after resting for 28 days. The designed electrolyte paves the way to develop high-voltage Zn batteries based on reversible lithiated cathodes.

Batteries based on Zn anodes offer a solution for safe, low cost and environmentally friendly energy storage[1,2]. However, the redox potentials of Zn (−0.76 V vs. SHE) are much higher than the Li electrode (−3.04 V vs. SHE), thus zinc batteries (ZBs) often operate at a lower voltage with low energy density than lithium-ion batteries (LIBs)[3,4]. Challenges to increasing the voltages of ZBs are twofold. First, aqueous electrolytes tend to decompose at a high voltage[5]. Some advanced strategies, including water-in-salt electrolytes[6] and the pH-decoupling electrolyte[7] have been developed to suppress the decomposition of aqueous electrolytes. Second, even though the electrochemical stability windows of electrolytes are expanded, high-potential cathode materials for ZBs are lacking[8]. By far, the benchmark materials, Prussian blue analogues (PBA), show a discharge plateau voltage of about 1.7 V (vs. $Zn^{2+}$/Zn), which is still far below the voltage of LIBs[9].

High-potential cathode materials for LIBs provide a large selection of suitable materials for ZBs[10]. For example, the potential of spinel $LiNi_{0.5}Mn_{1.5}O_4$ (LNMO) reaches ~4.7 V vs. $Li^+$/Li, which can be translated to about 2.5 V vs. $Zn^{2+}$/Zn[11]. However, there is no report

to achieve the highly reversible Zn∥LNMO battery so far. For conventional Zn/Li hybrid batteries like Zn∥LiFePO₄ or ZnLiMn₂O₄ batteries, the electrolyte is the key component and normally is the direct mixture of Zn salt and Li salt in solvent or polymer[12,13]. Thus, we tried to directly pair LNMO cathode with the Zn metal anode based on the Zn/Li mixture electrolyte (Fig. 1a). As shown in Fig. 1b, the resulting Zn∥LNMO battery exhibits poor reversibility with low Coulombic efficiency (CE) (-20%). Then, we collected the XRD pattern of LNMO electrode after cycling, clear characteristic peaks of $ZnMn_2O_4$ can be detected, indicating that part of LNMO was irreversibly converted into $ZnMn_2O_4$ during cycling (Fig. 1c). In addition, we also collected the X-ray photoelectron spectroscopy (XPS) of the electrodes at the initial state and after cycling, an obvious signal of Zn 2p can be detected in the electrode after cycling, indicating the intercalation of $Zn^{2+}$ ions during cycling (Supplementary Fig. 1). Furthermore, the high-resolution transmission electron microscopy (HRTEM) images of the electrode after cycling also confirm the existence of $ZnMn_2O_4$ (Supplementary Fig. 2).

[1]Department of Materials Science and Engineering, City University of Hong Kong, 83 Tat Chee Avenue, Kowloon, Hong Kong 999077, China. [2]Research Center for Materials, Architectures, and Integration of Nanomembranes (MAIN), TU Chemnitz, 09126 Chemnitz, Germany. [3]Material Systems for Nanoelectronics, TU Chemnitz, 09107, Chemnitz, Germany, TU Chemnitz, 09126 Chemnitz, Germany. [4]Hong Kong Center for Cerebro-Cardiovascular Health Engineering (COCHE), Shatin, NTHKSARChina. [5]Hong Kong Institute for Clean Energy, City University of Hong Kong, Kowloon 999077, Hong Kong. [6]These authors contributed equally: Ze Chen, Tairan Wang. ✉e-mail: minshen.zhu@main.tu-chemnitz.de; junfan@cityu.edu.hk; cy.zhi@cityu.edu.hk

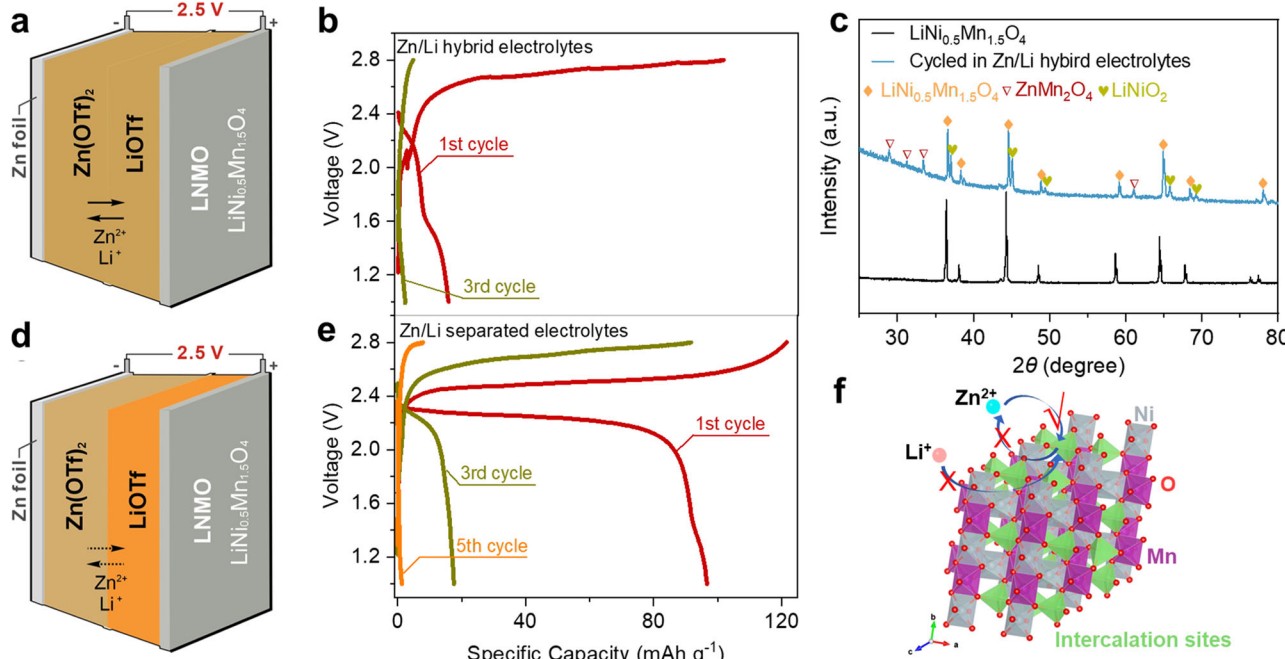

**Fig. 1 | Challenges to fabricate a rechargeable Zn‖LNMO battery. a** Schematic illustration and **b** the galvanostatic charge/discharge (GCD) curves of Zn‖LNMO battery based on the mixture electrolytes of zinc trifluoromethanesulfonate ($Zn(OTF)_2$) and lithium trifluoromethanesulfonate (LiOTF); **c** XRD patterns of the LNMO electrode at the initial state and discharging state (1.0 V) after cycling; **d** Schematic illustration and **e** the GCD curves of Zn‖LNMO battery based on separated $Zn^{2+}$ and $Li^+$ polymer electrolytes; **f** Spinel structure of LNMO with $Zn^{2+}/Li^+$ insertion in the hybrid electrolytes.

It is worth noting that $ZnMn_2O_4$ normally exhibits notoriously poor electrochemical activity in non-aqueous systems[14], $Zn^{2+}$ ions will be immobilized once intercalated into the LNMO structure. The irreversible intercalation of $Zn^{2+}$ ions is even more favorable than $Li^+$ ions[15]. In addition, obvious characteristic peaks of $LiNiO_2$ are detected, suggesting the incomplete reduction of $Ni^{4+}$ in the discharging process, further confirming the blocked $Li^+$ ion insertion due to the generation of $ZnMn_2O_4$[16]. Therefore, it is essential to avoid the emergence of $Zn^{2+}$ at the cathode side. Inspired by the phase separation electrolyte[17], we tried a simple separation of $Zn^{2+}$ ions and $Li^+$ ions in the electrolyte by using two polymer membranes (Fig. 1d), which turned out to be not very helpful. Although the separated solid polymer electrolytes successfully lead to the first charge/discharge cycle to normal operation, the battery capacity decays to near zero within only 5 cycles (Fig. 1e). Such a phenomenon can be attributed to the shutting of $Zn^{2+}$ ions to the cathode side, indicating the simple separation of $Zn^{2+}$ and $Li^+$ ions by two polymer membranes without additional design is inefficient in constructing reversible Zn‖LNMO battery[18]. Therefore, the main challenge to high-voltage Zn‖LNMO batteries relies on a delicate electrolyte configuration impeding the $Zn^{2+}$ ion shuttling and intercalating into LNMO during cycling (Fig. 1f).

Herein, we develop a polymer hetero-electrolyte (PHE) comprising two different polymer layers for the operation of the anode and cathode separately. The anode polymer layer (APL) allows for fast $Zn^{2+}$ ion transport, while the cathode polymer layer (CPL) blocks the $Zn^{2+}$ ion transport with the addition of crosslinked poly (methyl acrylate) (PMA). The blocking effect stems from the strong coordination of carbonyl oxygen atoms in PMA with $Zn^{2+}$ ions. The immobilized $Zn^{2+}$ ions also form a long-range structure with anions, resulting in a low $Zn^{2+}$ conduction. As such, a $Zn^{2+}$-blocking layer is established at the CPL and APL interface, preventing the $Zn^{2+}$ ion crossover. The PHE enables the operation of rechargeable Zn‖LNMO batteries, which exhibit a discharge plateau voltage of 2.4 V, and stable cycling performance with high CE.

## Results

### Fabrication, morphologies, and structures of PHE

Figure 2a illustrates the structure of PHE. Poly(vinylidene fluoride-co-hexafluoropropylene) (PVHF), a well-known polymer with high mechanical and chemical stability but a low crystallinity, is used as the matrix for both APL and CPL[19]. Succinonitrile (SN) is a plasticizer to enhance the ionic conductivity[20]. For the CPL, crosslinked methacryl polyhedral oligomeric silsesquioxane (POSS)-PMA is added to suppress $Zn^{2+}$ ion shuttle and improve the electrochemical stability at high voltages. Both APL and CPL are homogeneous, evidenced by the smooth surface (Fig. 2b, c). Furthermore, the smooth surfaces of PHE can be well maintained with a large-scale size, demonstrating the potential of scale-up (Supplementary Fig. 3). The PHE thickness is measured as about 49 μm, and more importantly, a homogeneous but clearly delimited interface is observed (Fig. 2d). As expected, the $Zn^{2+}$ and $Li^+$ are confirmed to be completely separated by the element mapping as shown in Fig. 2e, f, evidencing the successful preparation of the PHE. Note that the CPL thickness is three times that of APL, which physically increases the migration distance of $Zn^{2+}$ ions to LNMO. Thermogravimetric analysis (TGA) results show the main decomposition temperature ($T_d$) at around 332 °C for all CPL, APL, and PHE, which is attributed to the decomposition of the polymer matrix (Supplementary Fig. 4)[20]. In addition, before 150 °C, there is around 8.8% weight loss, which can be attributed to the residual water and solvent during test[21]. As shown in the differential scanning calorimetry (DSC) test, the molten temperature of PVHF is 144 °C for APL (Supplementary Fig. 5). With the addition of crosslinked POSS-PMA, the chain stability of PVHF increases. As a result, the PVHF molten temperature of CPL and PHE increases by 6.1 °C. Mechanical performance test (Supplementary Fig. 6) demonstrates that CPL is relatively brittle, and APL is ductile. PHE lies in the middle of CPL and APL, indicating the inherited polymer chain dynamics and mechanical properties in PHE. Moreover, due to the crosslinking network constructed by the POSS-PMA with a large number of inorganic components, PHE also shows

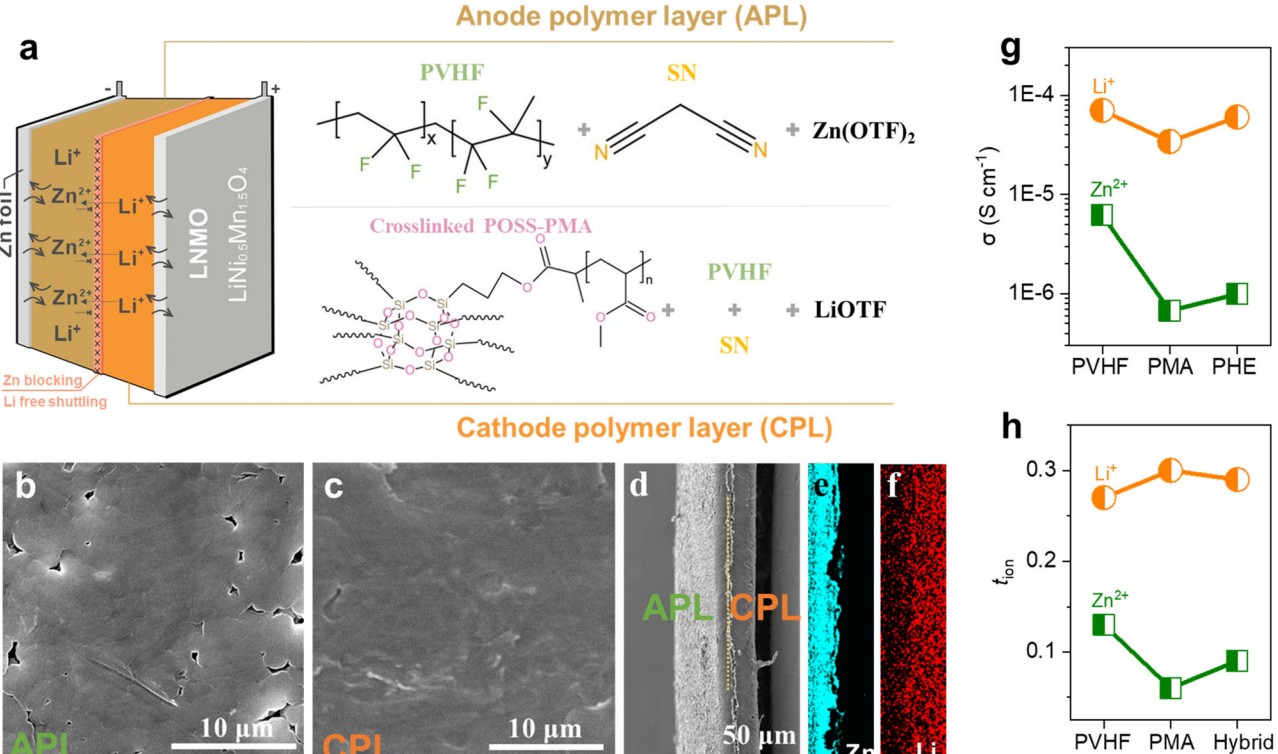

**Fig. 2 | PHE composition and ion mobility. a** Schematic illustration of PHE used in a battery, and the right panel describes the components for CPL and APL. SEM image of **b** the surface of APL, **c** the surface of CPL, and **d** the cross-section of PHE.

Elemental mappings of the PHE: **e** Zn and **f** Li. **g** Ionic conductivities (σ) and **h** ion transfer number of $Zn^{2+}$ and $Li^+$ ions in different polymer matrix.

fire-retardant capability, thus improving the safety of the cell operation (Supplementary Fig. 7)[22].

Furthermore, aiming at achieving the successful operation of the Zn‖LNMO cell, ion migration in different polymer matrices was investigated. The ionic conductivity of $Zn^{2+}$ ions in PMA is $5.9 \times 10^{-7}$ S cm$^{-1}$, which is far below the ionic conductivity of $Li^+$ ions in PMA ($3.5 \times 10^{-5}$ S cm$^{-1}$), confirming the sluggish migration kinetics of $Zn^{2+}$ ions in PMA matrix. Then, with the addition of PMA to the PVHF matrix, the conductivity of $Li^+$ ions in PHE slightly reduce from $7.06 \times 10^{-5}$ to $5.89 \times 10^{-5}$ S cm$^{-1}$. By contrast, $Zn^{2+}$ ion conductivity decreases vastly to $8.7 \times 10^{-7}$ S cm$^{-1}$, indicating that the $Zn^{2+}$ ion conduction is significantly suppressed after adding PMA to the polymer matrix (Fig. 2g). Similarly, compared with the ion transference number ($t_{ion}$) in PVHF matrix, $Li^+$ ion contribution ($t_{Li^+}$) in PHE remains stable and $Zn^{2+}$ ion contribution ($t_{Zn^{2+}}$) in PHE decreases dramatically due to the impeded ion mobility with the addition PMA (Fig. 2h). All the above results confirm the sluggish $Zn^{2+}$ diffusion resulting from the PMA, which indicates the successful suppression of the $Zn^{2+}$ ion shuttling in PHE.

**Mechanism of $Zn^{2+}$-blocking in PHE**

The reduced $Zn^{2+}$ ion conduction induced by PMA was investigated by molecular dynamic (MD) simulations. Figure 3a shows the snapshots of $Zn^{2+}$ and $Li^+$ ions in specific polymer electrolyte layers. Both cations ($Li^+$ and $Zn^{2+}$) coordinate with OTF$^-$ anions and the PMA chain. Oxygen atoms from OTF$^-$ anions and carbonyl groups in the PMA contribute to the first coordination shell of cations (Supplementary Fig. 8). The coordination number for oxygen atoms in OTF$^-$ anions is around 2.3 for both Zn and Li, indicating the OTF$^-$ anion has little effect on the change of ion conduction. The major difference arises from the coordination of the PMA chain. The coordination number for carbonyl oxygen atoms is 3.39 for $Zn^{2+}$ and it is 2.34 for $Li^+$ (Supplementary Fig. 8), which implies $Zn^{2+}$ ions are more encased in the polymer chain than for $Li^+$ ions. As such, the transport of $Zn^{2+}$ ions needs to overcome more

barriers, which can be further confirmed by the much higher binding energy of $Zn^{2+}$ ion pairs with PMA (Supplementary Fig. 9). Moreover, as shown in Supplementary Fig. 10, a long-range structure is observed for $Zn^{2+}$ coordinated with OTF$^-$ anions, which is also confirmed by the XRD results of clear peaks of Zn(OTF)$_2$ with the addition of PMA (Supplementary Fig. 11). The long-range structure further implies the high localization of $Zn^{2+}$ with PMA, which is totally different from the $Li^+$ ion in PMA matrix that no long-range coordination structure of $Li^+$ ion can be observed in Supplementary Fig. 8. In addition, obviously sluggish $Zn^{2+}$ diffusion trend can be observed in Supplementary Fig. 12, further confirming that the long-range coordination structure will obviously reduce the ion migration kinetics. In theory, the low diffusion coefficient of the $Zn^{2+}$ ion in PMA is more than five times lower than the $Li^+$ ion (Fig. 3b).

In general, such a double-layer polymer structure contributes an electrolyte resistance[23]. However, we found the ionic conductivity of PHE does not decrease compared to CPL and APL. As shown in Supplementary Fig. 13, the ionic conductivity of PHE can reach $1.48 \times 10^{-4}$ S cm$^{-1}$ at 25 °C, and it gradually increases with the temperature increasing, eventually reaching a maximum of $1.24 \times 10^{-3}$ S cm$^{-1}$ at 105 °C. The additional resistance is only exemplified at low temperatures (<5 °C) and the resistance increases as the temperature decreases (Supplementary Fig. 14). Nyquist plots of PHE under low temperatures demonstrate that the additional resistance stems from a charge transfer process. As no clear charge transfer resistance is found in CPL and APL at low temperatures (Supplementary Fig. 14), the additional charge transfer is attributed to the ion transport across the polymer-polymer interface[23–25]. Furthermore, the Bode plots of the PHE at temperatures ranging from −35 to 105 °C were employed to investigate the influence of the polymer-polymer interface on the whole ion transport resistance (Fig. 3c)[23]. Above 5 °C, the features belonging to the polymer-polymer interface become indistinguishable, indicating the negligible polymer-polymer interface resistance ($R_{pi}$), which was

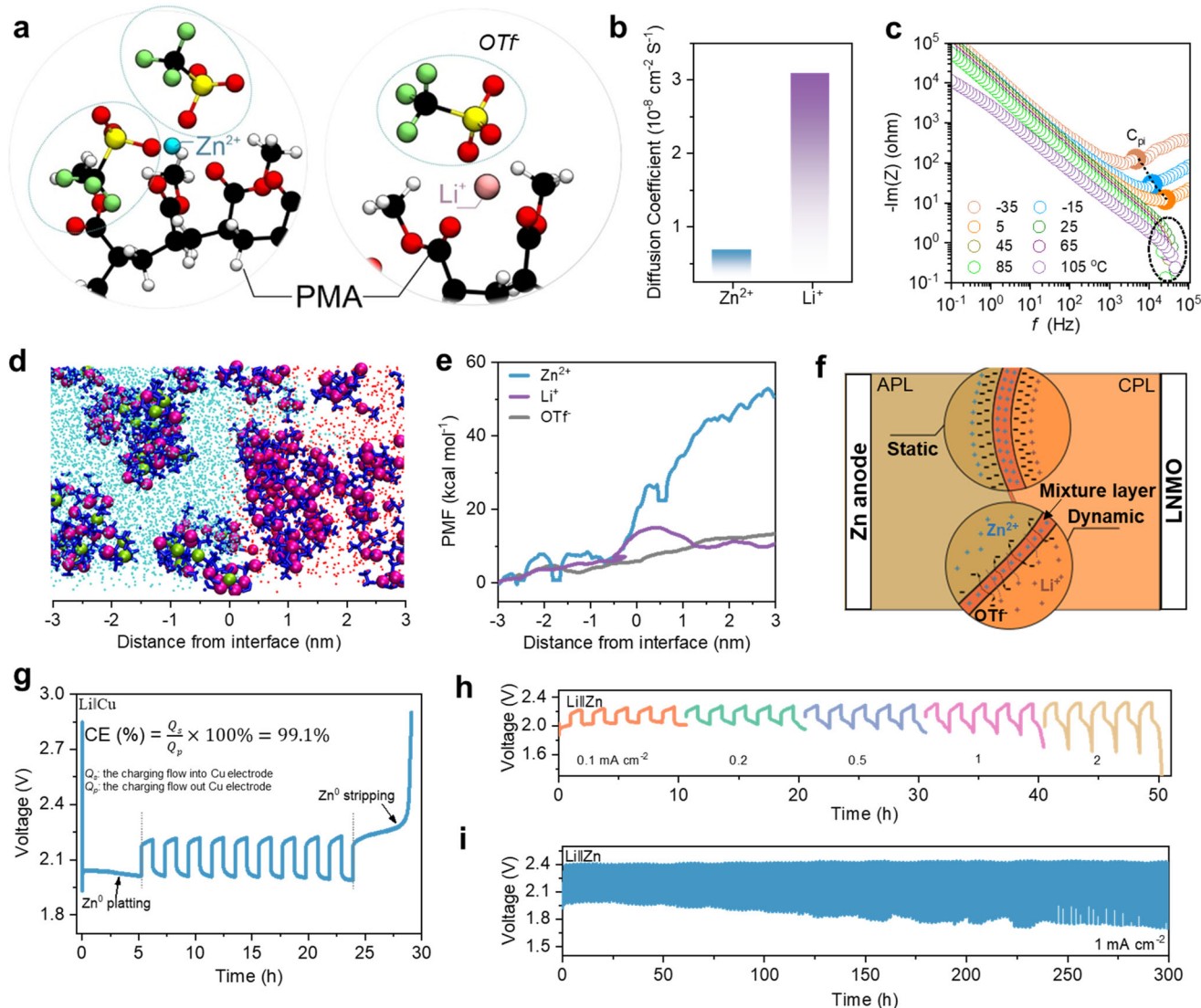

**Fig. 3 | Investigation of the ion migration in PHE. a** Snapshots of cations (Li[+] and Zn[2+]) and anion (OTF[-]) in PMA; **b** Diffusion coefficient of Zn[2+] and Li[+] in PMA matrix; **c** Bode plots of the PHE at different temperatures; **d** Simulation of ions migration in the PHE (green sphere: Zn[2+], purple sphere: Li[+], dark blue bond: OTf[-], light blue point: F atom from PVHF, red point: O atom from PMA); **e** PMF of different ions moving from APL to CPL side; **f** Schematic illustration of ions migration in the PHE; **g** GCD curves of Li‖Cu cell for the CE test of metal Zn electrode; Li‖Zn cell based on the PHE: **h** GCD curves at different current density and **i** cycling performance at 1 mA cm[−2].

also confirmed by the equivalent circuits based on the PHE configurations (Supplementary Fig. 15)[23,24]. The main resistance to ion transport in the PHE is from the bulk matrix ($R_{CPL + APL}$) rather than the $R_{PI}$, indicating that the PHE has great potential to achieve stable Li[+] transport across the polymer-polymer interface (above 5 °C). It should be emphasized that the bidirectional transport of Li[+] at the interface is of primary importance, which ensures the uniform charge distribution in the electrolyte.

Furthermore, we investigated the ion migration across the polymer-polymer interface based on MD simulation (Fig. 3d). During the ion migration process, Li[+] and OTF[-] ions can freely shuttle through the polymer-polymer interface, but Zn[2+] ion is strictly trapped at the interface between APL and CPL. The energy barrier of the ion migrating across the interface can further confirm the above description (Fig. 3e), in which the potential of mean force (PMF) of Zn[2+] ion moving across the interface vastly increases compared with Li[+] and OTF[-] ions and the PMF of Zn[2+] ion shutting to CPL side is 52.8 kcal mol[−1]. Based on the resistance test and MD simulations, we proposed the possible mechanism of the PHE (Fig. 3f). During the discharging process, a

mixture layer will form on the polymer-polymer interface, which mainly consists of the Zn salt with PMA from CPL. Such a mixture layer inhibits the further diffusion of Zn[2+] ions from the APL side and allows the free shuttling of Li[+] and OTf[-] ions in the PHE. During the charging process, the Zn[2+] ions in the mixture layer also own the trend to migrate back to the APL side though the sluggish kinetic of Zn[2+] migration in such layer reduces the efficiency of Zn[2+] ion, which at least promises the relative stability of the mixture layer with a dynamic Zn[2+] ion concentration equilibrium.

Furthermore, we explored the electrochemical stability and reversibility of Zn and Li metal anode based on the as-prepared PHE. As shown in Supplementary Fig. 16, the PHE is stable up to 2.14 V (vs. SHE), equaling 2.9 V vs. Zn/Zn[2+]. Such a high stable voltage demonstrates a remarkably enhanced high voltage tolerance by POSS-PMA crosslinking networks (Supplementary Fig. 17)[5]. In addition, a Li‖Cu cell was assembled to investigate the Coulombic efficiency (CE) of metal Zn based on the PHE (Fig. 3g). The Li platting/stripping and Zn platting/stripping occur at the Li side and Cu side, respectively and the CE of metal Zn is ~99.1%, indicating the high stability and high efficiency of

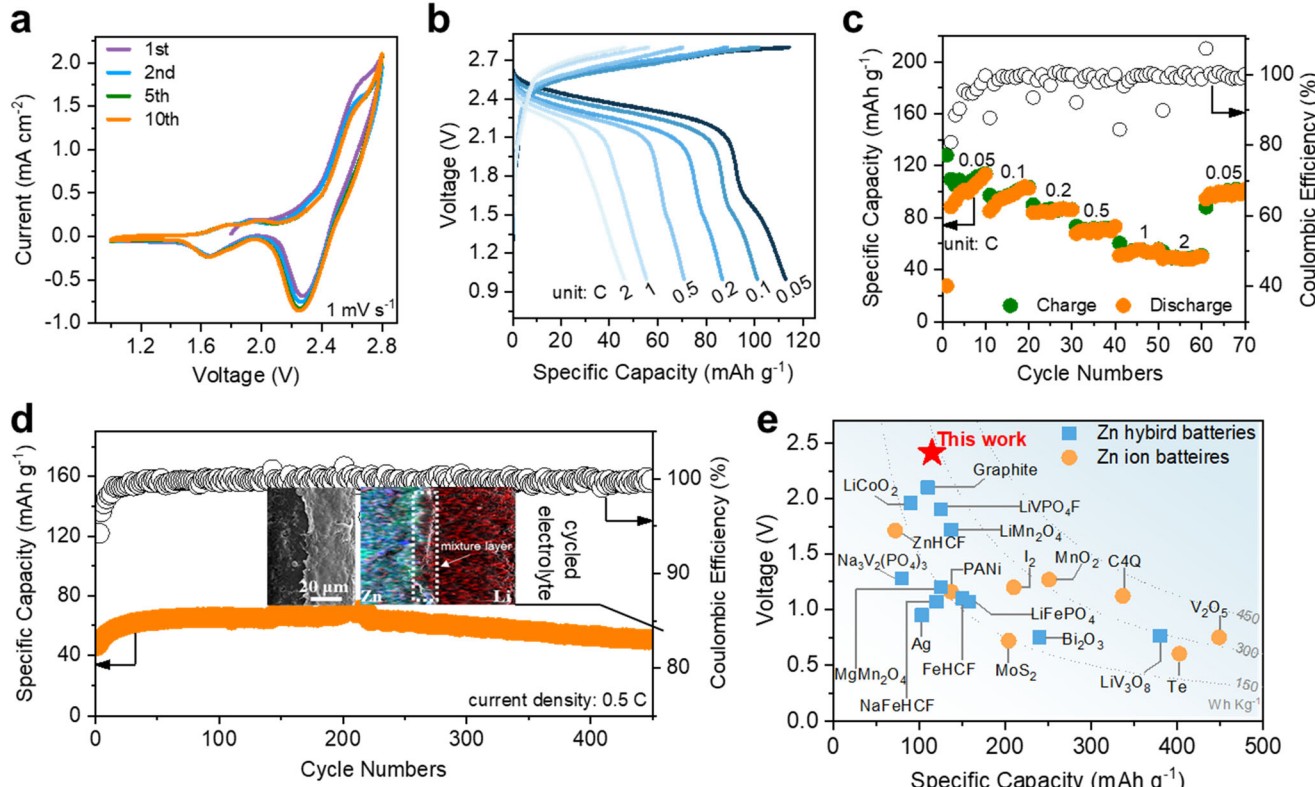

**Fig. 4 | Electrochemical performance of the Zn‖LNMO coin cell. a** CV curves at 1 mV s⁻¹; **b** GCD curves and **c** the rate performance at different rates; **d** Cycling performance and Coulombic efficiency at 0.5 C (the inset picture shows the SEM image and element mappings of the PHE after cycling (green dots: Zn²⁺, red dots: Li⁺)); **e** Comparison of the discharge voltage and capacity of Zn ion batteries and Zn hybrid batteries at room temperature (PANi: polyaniline; ZnHCF: zinc hexacyanoferrate; C4Q: calix[4]quinone; NaFeHCF: sodium-iron hexacyanoferrate; FeHCF: iron hexacyanoferrate.).

PHE. We also investigated the platting/stripping performance of Zn²⁺ ions with different capacities, which is shown in Supplementary Fig. 18. Stable platting/stripping of Zn²⁺ ions can be maintained when the capacity is below 3 mAh cm⁻² with high CE (above 95.1%). Moreover, as shown in Fig. 3h, stable Zn/Li deposition/dissolution with low polarization voltage in Li‖Zn cell can be maintained even at 2 mA cm⁻². Then, stable 300 h cycling of Li‖Zn cells at high current density (1 mA cm⁻²) can also be achieved, further confirming the advance of the PHE design. In addition, the $R_i$ of the Li‖Zn cell shows an obvious variation after cycling (from 574 Ω to 463 Ω), confirming the enhanced interface compatibility between the electrode and the PHE during cycling (Supplementary Fig. 19)[26].

### Electrochemical performance of zinc batteries with PHE

The practicability of PHE is evaluated in a coin cell using a Zn foil as the anode, and LNMO as the cathode (Zn‖LNMO cell). CV curves (Fig. 4a) show two-phase redox reactions and the redox peaks at over 2.5 V are attributed to the Ni²⁺/Ni³⁺/Ni⁴⁺ redox couple. Below 2 V, the redox couple of Mn³⁺/Mn⁴⁺ is also observed. As shown in the GCD curves of the Zn‖LNMO battery (Fig. 4b), at a low rate (0.05 C, 1 C = 145 mA/g), a long (~2.41 V) and a short (~1.53 V) discharge plateau also indicates the two-phase redox reactions. From both CV and GCD results, it is clear that the capacity contribution from the Mn³⁺/Mn⁴⁺ redox couple is much less than the Ni²⁺/Ni³⁺/Ni⁴⁺ redox couple. Normally, after the complete Ni redox in LNMO, it is hard for Mn to be further reduced because there is no free site for Li⁺ to be inserted into the spinel structure of LNMO[27]. Upon increasing the rate, the discharge plateau of the Mn³⁺/Mn⁴⁺ redox couple disappears due to its kinetic limitation. However, the discharge plateau at about 2.4 V maintains well even when the rate reaches 2 C (Fig. 4b), demonstrating the fast reaction kinetics of the Ni²⁺/Ni³⁺/Ni⁴⁺ redox couple. In specific, the Zn‖LNMO cell

delivers capacities of 113, 101, 86, 71, 55, and 45 mAh g⁻¹ at rates of 0.05, 0.1, 0.2, 0.5, 1, and 2 C, respectively (Fig. 4c). The Zn‖LNMO cell can cycle for 450 cycles at 0.5 C with a capacity up to 70.1 mAh g⁻¹ (Fig. 4d). The capacity at the initial cycles exhibits an increasing trend, which can be ascribed to the improved compatibility between the electrode and polymer electrolyte[11]. The capacity retention after 450 cycles is ~77.3% and the CE is maintained above 99.6%. Furthermore, we compared the cycling performance and CE of our battery with other Zn hybrid batteries (Supplementary Table 1), which further confirms the great superiority of the solid Zn‖LNMO cell to other systems. The inset picture in Fig. 4d shows the SEM image and EDS mapping of the PHE after cycling, in which the hetero-layer structure is still observed. More importantly, the absence of Zn in the CPL demonstrates that the addition of PMA can effectively block the Zn movement to the cathode. In addition to LNMO, characteristic redox peaks (Supplementary Fig. 20) and discharge plateaus (Supplementary Fig. 21) for LiCoO₂ and NCM811 are observed by using PHE as the electrolyte and Zn as the anode[28]. Such results demonstrate the versatility of PHE in cells using cathode materials for lithium-ion batteries but with Zn instead of Li as the anode.

Figure 4e compares the voltage and capacity of batteries using Zn as the anode. 2.0 V is a ceiling for most Zn batteries[29]. Though Zn‖Graphite battery owns ~2.1 V output voltage, the battery is mainly operated in the organic system for achieving stable cycling, leading to safety concerns[30]. Some other cathodes, such as LiVPO₄F, ZnHCF, and LiMn₂O₄, is not impeded by Zn²⁺ ion intercalation, they can be directly used in Zn hybrid batteries[31]; however, their operation voltages have not surpassed 2.0 V due to the limit of their intrinsic redox potential[32,33]. Significantly, with PHE, the Zn‖LNMO cell breaks the limit of 2.0 V, reaching 2.41 V. The significantly improved voltage leads to a high energy density though the capacity is lower than common

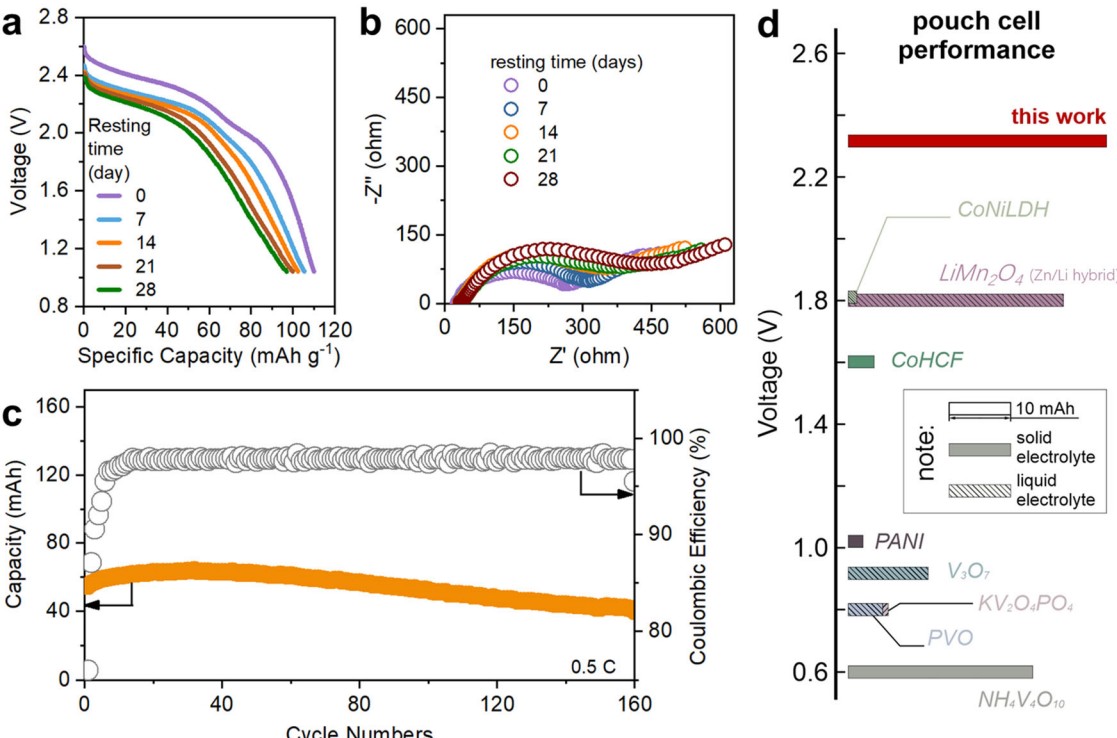

**Fig. 5 | Electrochemical performance of the Zn‖LNMO pouch cell. a** GCD curves and **b** Nyquist plots of the solid Zn‖LNMO battery after resting. **c** Cycling performance of the solid Zn‖LNMO pouch cell at 0.5 C; **d** Comparison of pouch cell performance to date.

high-capacity cathodes (e.g., $MnO_2$ and $V_2O_5$) for Zn batteries with operation voltage below 1.5 V[34,35]. As shown in Fig. 4e and Supplementary Table 1, the energy density of the Zn‖LNMO cell is projected to ~300 Wh kg$^{-1}$, approaching the first-tier energy density of Zn ion batteries and Zn hybrid batteries[30,32,34,36–52].

### Zn‖LNMO pouch cell

In order to evaluate the PHE performance in a more practical scenario, we assembled Zn‖LNMO pouch cells with a maximum capacity of 40 mAh. The CV curve of the pouch cell is consistent with the results of the coin cell, indicating the reliability during the scale-up of the cell (Supplementary Fig. 22). Figure 5a shows the GCD curves of the pouch cell, demonstrating the characteristic discharge plateau at about 2.44 V. At the rate of 0.05 C, the specific capacity is 112 mAh g$^{-1}$. As the rate increases to 0.5 C, the specific capacity decreases to 63 mAh g$^{-1}$ (Supplementary Fig. 23). It is worth noting that the capacity of the pouch cell is slightly lower than that of the coin cell, which can be attributed to the decreased ion diffusion kinetic in the thick electrode of pouch cell[53]. These results indicate scaling up is feasible for the Zn‖LNMO cell with PHE as the electrolyte.

Besides the dynamic discharge performance, the ability to inhibit the self-discharge of cells is crucial for deployment in realistic applications[53]. As shown in Fig. 5a, the plateau voltage slightly decreases to 2.3 V after resting for 7 days, and at the same time, the capacity drops to 105 mAh g$^{-1}$. The voltage and capacity of the pouch cell are relatively stable over the prolonged resting time of 28 days. High-capacity retention of around 88% with an average discharge voltage of 2.2 V was achieved after resting. Figure 5b shows the Nyquist plots of the pouch cell during the resting tests. The charge transfer resistance increases as the resting time grows, which can be attributed to the accumulation of Zn ions at the interface of CPL and APL. The Zn‖LNMO pouch cell can cycle 160 times with a capacity retention of about 77%. The CE after initial cycles is stabilized at over 95%. The energy density of the pouch cell is ~142 Wh L$^{-1}$ calculated based on the

anode, cathode, and PHE, and ~54 Wh L$^{-1}$ calculated based on the whole cell (Supplementary Table 2). Based on such good performance, one single Zn‖LNMO pouch cell can light up an LED display, and owing to the good mechanical properties of PHE, the pouch cell can withstand deformation and cut (Supplementary Fig. 24). Figure 5d summarizes the pouch cell performance of Zn hybrid batteries, including batteries based on both polymer and liquid electrolytes[54–61]. PHE allows for the design of the Zn‖LNMO pouch cell with the highest voltage to date (2.4 V). Such a high voltage is achieved without sacrificing capacity. The pouch cell capacity is also the highest for Zn batteries using polymer electrolytes. The combined high voltage and high capacity nicely demonstrate the feasibility of PHE in scale-up batteries[62].

## Discussion

The high-potential lithiated cathodes based on reversible Li$^+$ ion intercalation are highly promising for a high-voltage zinc battery. Unfortunately, when Zn$^{2+}$ ions exist, the lithiated cathode prefers to intercalate Zn$^{2+}$ instead of Li$^+$ ions, followed by immobilization of Zn$^{2+}$ ions and poor reversibility. We developed a PHE electrolyte consisting of two polymer electrolytes (APL and CPL) to solve this problem. Due to the strong coordination of Zn$^{2+}$ with carbonyl oxygen atoms in PMA chains and long-range structure connected to OTF$^-$ anions, Zn$^{2+}$ ions were efficiently immobilized at the anode side polyelectrolyte. The PHE shows an ionic conductivity of $1.24 \times 10^{-3}$ S cm$^{-1}$ and allows separated Zn and Li platting/stripping (300 h cycling) with high reversibility in unique Li‖Zn cell. Furthermore, the Zn‖LNMO cell based on the PHE achieves reversible Li$^+$ intercalation for LNMO, which offers a discharge plateau at around 2.4 V, breaking the ceiling of Zn batteries (2 V). The battery not only shows good cycling stability and high energy density in a coin cell but also demonstrates the feasibility for the scale-up. The pouch cell also shows high stability with a large capacity of 40 mAh. The concept of inhibiting Zn$^{2+}$ ions from reaching the cathode proved to be feasible and the PHE electrolyte provided the prototype for using this method.

## Methods

### Materials

Poly(vinylidene fluoride-co-hexafluoropropylene) (Mw ~400,000) is purchased from Sigma Aldrich. Succinonitrile (SN), N, N-Dimethylformamide (DMF), lithium trifluoromethanesulfonate (LiOTF), Zinc trifluoromethanesulfonate (Zn(OTF)$_2$), Methyl acrylate (MA) and 2,2′-Azobis(2-methylpropionitrile) (AIBN) are purchased from Aladdin. MA was vacuum-distilled to remove the inhibitor. AIBN was recrystallized before use. Methacryl polyhedral oligomeric silsesquioxane cage mixture (POSS) was purchased from Forsman.

### Preparation of cathode polymer layer (CPL)

First, PVHF, SN, and LiOTF were mixed with a weight ratio of 75: 10: 15 in DMF. The suspension was stirred at 60 °C for 3 h. Then, MA (60 wt% of PVHF), POSS (1.5 mol% of MA), and AIBN (0.7 wt% of MA) were successively added to the above solution. The mixture solution was stirred and degassed with nitrogen for 20 min, then sealed in two glass plates, followed by heating at 65 °C for 4 h. The resulting membrane was further dried under vacuum at 70 °C for 48 h, and then at 120 °C for 6 h under vacuum. Next, the resulting films were placed between two PTFE sheets and heated to 160 °C for 15 min in a hydraulic lamination hot press. Finally, the CPL was obtained and stored in the glove box.

### Preparation of anode polymer layer (APL)

PVHF, SN, and Zn(OTF)$_2$ were mixed with a weight ratio of 90: 5: 5 in DMF. Then, the mixed solution was cast onto a glass plate and further dried at 60 °C under vacuum for 48 h, then at 120 °C for 6 h.

### Preparation of polymer hetero-electrolyte (PHE)

PHE was assembled by placing APL and CPL between two PTFE sheets and heating at 150 °C for 10 min followed by pressing at 1.5 MPa for 20 mins. Once again, the pressure was maintained until the temperature was cooled to 40 °C after the heater had been turned off. By this method, PHEs with a well-defined interface with no gaps were prepared.

### Characterization and Electrochemical measurements

Morphologies of the PHE and the electrodes were studied by the transmission electron microscope (FEI Tecnai F20) and the scanning electron microscope (SEM, FEI Quanta 450 FEG). The mechanical performance of the PHE was investigated by the tensile test (CMT6103). Zn foil was used as the anode electrode (20 μm), LiCoO$_2$, NCM811, and LNMO on Al foil with 1.5 mg cm$^{-2}$ (coin cell) and 8 mg cm$^{-2}$ (pouch cell) loading mass were used as the cathode. To enhance the compatibility of the electrode and the PHE, the as-fabricated solid batteries were stored at 110 °C for 2 h before the test. LAND CT2001A device and electrochemical workstation CHI 760D were used to collect the electrochemical properties of batteries.

### Computational simulation

A large-scale atomic/molecular parallel simulator (LAMMPS) package is used for the simulation[63]. The Amber-formed force field of the polymer is generated by the AmberTool. The atomic charge of the polymer is fitted by the Restrained Electrostatic Potential (RESP) on the level of HF/6-31G* in Guassian09. The electrolyte is firstly annealed from 800 to 300 K for 5 ns. Then, the system maintains at 300 K for 5 ns to reach equilibrium. Finally, another 5 ns production run is used for collecting the trajectory of molecules; the snapshots of trajectory are saved every 5 ps. The system is in the isothermal-isobaric ensemble (NPT) for all simulations. The mean square displacement of atoms is calculated by the Einstein Eq. (1):

$$MSD \equiv \sum_{i}^{N} \left\langle \left( R_i(t) - R_0(t) \right)^2 \right\rangle \qquad (1)$$

where $R$ is the atomic coordinates, $N$ is total number of atoms, and $t$ is the time. The diffusion constant ($D$) is the slope of MSD versus times with a factor of 1/6, and the corresponding Eq. (2) is:

$$D = \frac{MSD}{6t} \qquad (2)$$

The Vienna Ab initio Simulation Package (VASP)[64] is used for density functional theory calculations. The force-based criterion of structural optimization and CI-NEB are set to 0.01 eV/Å.

The potential of mean force (PMF) simulations were performed with GROMACS package with the umbrella sampling method and the gmx wham tool in GROMACS. The interval of umbrella sampling windows is 1.0 Å. In each window, the distance of the Li, Zn, and OTF ions along the z-axis was restrained with a harmonic force constant of 1000 kJ mol$^{-1}$ nm$^{-2}$. Each window was simulated for 20 ns.

### Reporting summary

Further information on research design is available in the Nature Portfolio Reporting Summary linked to this article.

## Data availability

The data that support the findings of this study are available within the text including the Methods, and Supplemental information. Raw datasets related to the current work are available from the corresponding author upon reasonable request. Source data are provided in this paper.

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

## Acknowledgements

This research was supported by the National Key R&D Program of China under Project 2019YFA0705104. This research was also supported by RGC Collaborative Research Fund under Project C1002-21G and in part by InnoHK Project on [Project 1.4 - Flexible and Stretchable Technologies (FAST) for monitoring of CVD risk factors: Soft Battery and self-powered, flexible medical devices] at Hong Kong Centre for Cerebro-cardiovascular Health Engineering (COCHE). The work was also partially supported by a grant from the Research Grants Council of the Hong Kong Special Administrative Region, China (Project No. R5019-22).

## Author contributions

C.Y.Z. conceived the project. M.S.Z. and C.Y.Z. supervised the research. Z.C., Z.X.W. and Y.H. prepared the materials. Z.C., A.C., Y.B.W. and Z.D.H. conducted the characterization and analyzed the data. T.R.W. and J.F. performed molecular simulation. Z.C., O.S., M.S.Z. and C.Y.Z. wrote the paper and all authors engaged in discussions related to the manuscript.

## Competing interests

The authors declare no competing interests.
