## [Peer Review File · Nature Communications]

Polymer Hetero-Electrolyte Enabled Solid-state 2.4-V Zn/Li Hybrid BatteriesREVIEWER COMMENTS

Reviewer #1 (Remarks to the Author):

The major problem that this paper aims to address is the insertion of Zn ions to LNMO, which authors claim will block subsequent Li ion insertion and results in irreversibility, but no evidence was supported to confirm Zn ions can indeed be inert to LNMO and such surface layers can indeed block Li ion insertion, as such, the major novelty of this paper appears broadly unsupported. Some other reasons may cause the irreversibility and degradation reported in Figure 1. Some other areas that the authors may want to consider for improvements are: 1) hybrid electrolytes with multiple charge carriers are widely explored in the literature and the technical problems, especially with low energy density due to the need for much higher electrolyte volumes, are also widely recognized, I do not see how authors address this issue and as such, discussions regarding energy densities are ungrounded; 2) the charge transport mechanism across the two polymer layers is very elusive, it is unclear how it sustains for high capacity and long cycling, the discussion of charge transfer resistance on page 9 requires more evidence, charge transfer from what reactions exactly? 3) I do not quite understand Figure 3h and i, to be convincing authors should provide stabilities at much higher capacities, 4) literature typically doesn't discharge LNMO to low voltages so that to avoid Mn redox, but in Figure 4b the Mn redox is unclear (it should have roughly equivalent capacity as the Ni redox). 5) the stability of PHE and dynamics of the interface are essentially missing, and as such, it is unclear if the functional mechanisms proposed by the authors are valid.

Reviewer #2 (Remarks to the Author):

This manuscript reported polymer hetero-electrolytes for solid Zn batteries, which successfully block the Zn²⁺ shuttle and allow the separated Zn and Li reversibility. Benefiting from the unique polymer electrolyte, high-voltage lithium cathode is able to work in Zn/Li hybrid batteries. The as-resulted solid Zn/Li hybrid batteries can deliver significantly high output voltage compared with conventional zinc ion batteries, accompanying with the long cycling life and the great potential for large-scale application. The work is inspiring for constructing multi-ion hybrid system with tailored ion diffusion, and the strategy is potentially applicable for other metal ion batteries for achieving high-voltage and high-energy output. Thus, I recommend publishing this work in Nature Communications. Some comments are as followed:

1. As solid electrolytes, the interface between the electrode and solid electrolytes cannot be ignored and it has a huge influence on the electrochemistry performance. Normally, some special treatments are necessary. How do the authors overcome this problem? Detailed procedures should be added in the experimental section.
2. High-voltage-type lithium cathode LNMO was employed in the hybrid battery and the battery shows remarkable cycling performance based on the polymer electrolytes. How about the high-voltage tolerance of the polymer electrolyte. What are the key factors affecting the high-voltage resistance of polymer electrolytes? It is due to the enhanced molecular interaction or the suppressed polymer chains kinetics in CPL?
3. The Zn-LNMO cell was fabricated based on the as-prepared polymer hetero-electrolyte. The CE gradually got stable at above 99.6 % during the cycling. The CE should be

compared with other Zn hybrid batteries reports to further reveal the decent ionic migration performance in the polymer electrolytes.

4. As shown in the Figure 4d and 5c, the cycling performance of LNMO exhibits obvious capacity increase at the beginning of cycling, what is the reason? For the high-voltage LNMO cathodes, what factors will result in the capacity decay during cycling, and whether the as-prepared polymer electrolytes can effectively suppress these issues?

5. As the authors stated in the manuscript, it seems that there are multiple strategies to block Zn^{2+} motion in cathode side, including the selection of different polymer matrix in CPL and APL, the different thickness of CPL and APL, and the ion pairs effect of Zn^{2+} and Li^+ . What is the most important factor for achieving the highly blocked Zn^{2+} mobility, more explanations should be added in the manuscript.

6. In Figure 3j, the interface impedance of the Zn-Li battery gradually decreases, then further increases with the proceeding of cycling. It seems some side reactions happened. Please clarify the phenomenon.

Reviewer #3 (Remarks to the Author):

This work reports on fabricating hybrid, dual-layer, polymer electrolytes for Zn-Li batteries. The anode polymer layer (APL) consists of poly(vinylidene fluoride-co-hexafluoropropylene) (PVHF), Succinonitrile (SN) and Zn salt. In the cathode polymer layer, methacryl POSS/methyl acrylate was added to PVHF/SN/LiOTF. The motivation is to solve the problem of Zn^{2+} immobilization in LNMO cathodes. The design rationale is that the cathode layer blocks Zn^{2+} shuttling due to the strong coordination between PMA and Zn^{2+} . Full cell tests were conducted to demonstrate the success of the design. Dual-layer SPE has been widely used in various secondary battery designs, and this one is another interesting application. This reviewer has the following questions about the electrolytes and battery design.

1. The goal is to solve the problem of immobilized Zn^{2+} in LNMO. However, when introducing crosslinked PMA, the Zn^{2+} ions are strongly coordinated with the polymer. This is basically equivalent to cation immobilization. Are Zn^{2+} ions still mobile in the battery? Judging from the extremely low conductivity and transference number, they seem immobile. If so, do we lose a significant amount of Zn into the electrolytes in every cycle? This could be a critical problem for the design. In the experiment, how thick was Zn foil? Would it work for thin or even anode-free design?

2. Along the same line, does the Zn^{2+} concentration in PMA increase with cycling? This also affects lithium concentration, conductivity etc.

3. How thick was each layer? At 120C, the film has a ~5% weight loss, which is considered the thermal stability limit. However, the sample processing was conducted at 150C. Based on TGA, there is ~10-20% weight loss already for the samples, which is way beyond the thermostability of the materials.

4. In Figure 4e, detailed conditions such as rates, temperature etc. should be given for a fair comparison.

Reviewer #4 (Remarks to the Author):

In this paper, the authors deal with the polymer electrolyte design for Zn/Li hybrid batteries. The area of Zn battery as an emerging technology seeks ample attention. The most important aspect that the authors want to conclude in this study is to demonstrate that the polymer hetero-electrolyte (PHE), consisting of an anode layer for Zn²⁺ ion carries and a cathode layer for blocking Zn²⁺ ion carries, enables the sufficient cycling of 2.4 V LiNi_{0.5}Mn_{1.5}O₄/Zn batteries. 40 mAh pouch cells are also prepared to convince the scale-up potential and low self-discharge property. Even though the research direction and the research problem are remarkably imperative, quite some flaws need to be addressed before the consideration of the manuscript for publication in this outstanding journal.

Some important remarks are:

1. Does any other paper report about LiNi_{0.5}Mn_{1.5}O₄/Zn batteries? As well as the polymer electrolyte for Zn/Li hybrid batteries? What is the most distinguish achievement here?
2. The authors have to clearly mention the precedents on the major milestones to probe out the cycling of LiNi_{0.5}Mn_{1.5}O₄/Zn batteries, pointing to the novelty of the present attempt. In the introduction part, the authors have to explain the different electrolyte design of Li/Zn hybrid batteries, indicating the novelty of PHE beyond the previous research.
3. The authors mentioned that "Simple separation of Zn²⁺ and Li⁺ ions by two polymer membranes is hard ..." Here, the authors didn't cite any literature or provide any specific explanation to reach the postulate.
4. What is the Coulombic efficiency of metallic Zn in the designed PHE?
5. Besides molecular dynamic simulation in Figure 3, is there other specific characterization to prove Zn²⁺ ions are more encased in the PMA chain than Li⁺ ions.
6. In figure 4d, the X axis should be "Specific capacity" rather than "Voltage".
7. As mentioned by the authors, "... the energy density is projected to around 300 Wh kg⁻¹ ..." Here, is the energy density only based on active material in anode/cathode side? A detailed calculation method should be provided in the SI. Besides Figure 4e, it also better to provide a comparison figure for energy density differences in different Zn (hybrid) batteries.
8. What is the main reasons for the initial capacity increase in both coin and pouch cells? (Figure 4d and 5c) Why is the capacity 10 mAh g⁻¹ less in pouch cells than coin cells under the same rate?
9. The overall style of presentation and quite some sentences in the manuscript has to be further modified.

Response to Reviewers

Dear Reviewers,

Thanks a lot for your constructive comments and suggestions about our manuscript entitled "Polymer Hetero-Electrolyte Enabled Solid-state 2.4-V Zn/Li Hybrid Batteries" (NCOMMS-23-15899), which we really appreciate. All comments are greatly valuable and helpful for improving the quality of our paper. We have studied your comments carefully and accordingly revised our manuscript in hope of addressing your concerns and meeting the high standards of *Nature Communications*. The comments were addressed point-by-point below and the related changes have been highlighted in the revised manuscript and supporting information.

To Reviewer #1:

The major problem that this paper aims to address is the insertion of Zn ions to LNMO, which authors claim will block subsequent Li ion insertion and results in irreversibility, but no evidence was supported to confirm Zn ions can indeed insert into LNMO and such surface layers can indeed block Li ion insertion, as such, the major novelty of this paper appears broadly unsupported. Some other reasons may cause the irreversibility and degradation reported in Figure 1.

Reply:

Thank you for the valuable comment. To clearly reveal the chemical structure change of the LNMO-based electrode in the Zn/Li hybrid electrolyte during cycling, we have collected the XRD patterns of the LNMO electrodes at the initial state and discharging state (1.0 V) after 10 cycles (Figure R1), which was newly added as Figure 1c in the revised manuscript. From the XRD patterns, new peaks located at $\sim 28.9^\circ$, 31.2° , 33.4° and 61.1° are detected, which can be ascribed to ZnMn_2O_4 (PDF#24-1133). It indicates that part of LNMO is irreversibly converted into ZnMn_2O_4 when the batteries are operated in the Zn/Li hybrid electrolyte. It is worth noting that ZnMn_2O_4

normally exhibits notoriously poor electrochemical activity in non-aqueous systems without proton assistance (*Joule* **2020**, *4*, 771-799). Thus Zn^{2+} ions will be immobilized once intercalated into the LNMO structure. The irreversible intercalation of Zn^{2+} ions is even more favorable than Li^+ ions, blocking the accessible vacancies for Li^+ ions. Prominent characteristic peaks of LiNiO_2 (PDF#89-3601) also can be detected at the discharging state, suggesting the incomplete reduction of Ni^{4+} in the discharging process, confirming the blocked Li^+ ion insertion due to the generation of ZnMn_2O_4 .

We are sorry for not providing a clear explanation of the deteriorated cycling performance in the previous version of the manuscript. We have followed your comment to add more discussions in the corresponding positions; please see highlighted parts on pages 3, 4 and 5 in the revised manuscript.

Figure R1. XRD patterns of the LNMO electrode at initial state and discharging state (1.0 V) after cycling.

Comment 1: Some other areas that the authors may want to consider for improvements are: hybrid electrolytes with multiple charge carriers are widely explored in the literature and the technical problems, especially with low energy density due to the need of much higher electrolyte volumes, are also widely recognized, I do not see how authors address this issue and as such, discussions regarding energy densities are ungrounded.

Reply:

Thank you for the insightful comment. In traditional hybrid batteries, a large amount of hybrid electrolytes is used, which decreases energy density and costs. For example, for common zinc ion batteries or Zn-based hybrid batteries, especially the typical Zn||LiMnO₂ hybrid aqueous battery, the adopted Zn anode is excessive with above 100 μm thickness and thick glass fiber separator (> 300 μm) is also employed (*J. Energy Storage* **2022**, *45*, 103744; *Energy Rev.* **2022**, *1*, 100005.) (Table R1). Thus, the resulting energy density of the battery is low, and other shortcomings, including the low loading mass of active materials and the narrow electrochemical stability window of aqueous electrolytes, further impair the energy output of Zn-based hybrid batteries.

However, reduced membrane thickness (~49 μm) (Figure 2d in the revised manuscript) can be achieved in our developed polymer hetero-electrolyte (PHE), benefiting from the good mechanical performance of the polymer electrolyte, which is far thinner than the common glass fiber separators. In addition, the PHE also owns high ionic conductivity and good stability, resulting in the stable plating/stripping of Zn anode with high efficiency (new Figure 3g in the revised manuscript). Thus, the increased loading mass of active materials (~8 mg cm⁻² in pouch cell) and reduced thickness of Zn foil (20 μm) can be achieved in our battery. Moreover, the widened electrolyte window (anodic stability above 2.9 V vs. Zn²⁺/Zn) in our PHE also ensures the stable cycling of high-voltage LNMO electrodes.

Benefiting from the above merits, our battery can deliver ~300 Wh Kg⁻¹ energy density (calculated based on the active materials) with 2.41 V output voltage, which surpasses all the Zn-based hybrid batteries (Figure R2 and Table R2). Furthermore, the fabricated Zn||LNMO pouch cell can also deliver ~142 Wh L⁻¹ energy density (calculated based on the anode, cathode and polymer electrolytes, and the details of the calculations were added as new Table S2 in the supporting information), which is superior to common Zn-based hybrid batteries (normally < 100 Wh L⁻¹ energy density) (*J. Energy Storage* **2022**, *45*, 103744.).

We have followed your comment to add more discussions in the corresponding positions; please see highlighted parts on pages 12, 13 and 14 in the revised manuscript

and pages 16 and 17 in the supporting information.

Table R1. Comparison of our Zn||LNMO battery with the typical Zn hybrid batteries.

System	Electrolyte	The thickness of Zn anode (μm)	Separator	Output voltage (V)	Energy Density _a (node+cathode+electrolyte) (Wh L^{-1})	Ref
Zn LiMnO ₂ or LiFePO ₄	Liquid electrolyte with Zn + Li salts	>100	Glass fiber (>300 μm)	< 1.7	< 100	1
Zn LNMO	PHE (~49 μm)	20	/	2.4	142	This work

Figure R2. Comparison of the discharge voltage and capacity of Zn ion batteries and Zn hybrid batteries at room temperature (PANi: polyaniline; ZnHCF: zinc hexacyanoferrate; C4Q: calix[4]quinone; NaFeHCF: sodium-iron hexacyanoferrate; FeHCF: iron hexacyanoferrate.).

Table R2. Comparison of the Zn hybrid batteries in different systems.

Cathodes	Electrolyte ^a	Capacity (mAh g ⁻¹)@Rate	Energy Density (Wh Kg ⁻¹)	Coulombic Efficiency (%)	Cycling life ^b	Ref
MgMn ₂ O ₄	1M MgSO ₄ + 1M ZnSO ₄	125(0.1 A g ⁻¹)	150	99.0	80% (500)@0.5 A g ⁻¹	2
LiV ₃ O ₈	LiOTf + 3M Zn(OTf) ₂	380(0.1 A g ⁻¹)	285	88.8	87% (4000)@5 A g ⁻¹	3
LiCoO ₂	Zn(OAc) ₂ + 4M LiOAc + NH ₃ H ₂ O	91(0.5 C)	173	98.7	95% (300)@2 C	4
FeHCF	3M KOTf + Zn(OTf) ₂	150(0.5 A g ⁻¹)	165	94.0	57% (1000)@1 A g ⁻¹	5
LiVPO ₄ F	21 M LiTFSI + Zn(OTf) ₂	125(0.1 A g ⁻¹)	237	95.0	87% (600)@1 A g ⁻¹	6
LiFePO ₄	1M LiOTf + Zn(OTf) ₂	158(0.5 C)	174	94.0	89% (100)@1 C	7
Na ₃ V ₂ (PO ₄) ₃	1M Ca(OTf) ₂ + Zn(OTf) ₂	81(1 C)	105	99.0	74% (1300)@20 C	8
LiMn ₂ O ₄	1m Zn(TFSI) + 20m LiTFSI	65(0.2 C)	112	99.9	85% (4000)@4 C	9
Ag	0.1M ZnCl ₂	104(1.5 A g ⁻¹)	99	99.0	93% (1300)@1 A g ⁻¹	10
Graphite	3m Zn(TFSI) ₂ in ethyl methyl carbonate	110(0.1 A g ⁻¹)	231	94.0	96% (50)@0.1 A g ⁻¹	11
Bi ₂ O ₃	6M KOH + 0.3M Zn(OAc) ₂	323(0.3 C)	245	98.2	49% (1000)@10 A g ⁻¹	12
LNi _{0.5} Mn _{1.5}	PHE	113(0.05)	289	99.6	77.3%	Thi

-
- a. All the electrolytes are aqueous except special illustration.
b. The percentage means capacity retention, and the number in brackets means cycle numbers.

Comment 2: the charge transport mechanism across the two polymer layers is very elusive, it is unclear how it sustain for high capacity and long cycling, the discussion of charge transfer resistance on page 9 requires more evidence, charge transfer from what reactions exactly?

Reply:

Thank you for the insightful comment. To investigate the ion migration in the PHE primarily through the polymer-polymer interface, we first tested the charge transfer resistance, and the corresponding Bode plots of the PHE from the resistance test were employed to investigate the influence of the polymer-polymer interface to the whole ion transport resistance (Figure 3c in the revised manuscript). Local maxima associated with the capacitive processes of the polymer-polymer interface (C_{pi}) have been marked in the Bode plots. Above 5 °C, the features belonging to the polymer-polymer interface become indistinguishable, indicating the negligible polymer-polymer interface resistance (R_{pi}). Using the Bode plots, the equivalent circuits to simulate the impedance-frequency response of the PHE configurations were constructed (Figure S13 in the supporting information). The main resistance to ion transport in the PHE is from the bulk matrix rather than the R_{pi} , confirming the stable Li^+ transport across the polymer-polymer interface as efficient charge carriers.

To further clarify the ion migration mechanism in the PHE, a molecular dynamic (MD) simulation was performed to investigate the ion migration across the polymer-polymer interface (Figure R3a). During the ion migration process, Li^+ and OTf⁻ can freely shuttle through the polymer-polymer interface, but Zn^{2+} is strictly trapped at the interface between APL and CPL. The energy barrier of the ion migrating across the interface can further confirm the above description (Figure R3b), in which the potential of mean force (PMF) of Zn^{2+} moving across the interface vastly increases compared

with Li^+ and OTf^- and the PMF of Zn^{2+} moving in CPL side is $52.8 \text{ kcal mol}^{-1}$. Based on the resistance test and MD simulation, we proposed the possible mechanism of our PHE (Figure R4). During the discharging process, a mixture layer will form on the polymer-polymer interface, mainly consisting of the Zn salt with polymethyl acrylate from the CPL side. Such a mixture layer inhibits the further diffusion of Zn^{2+} from the APL side and allows the free shuttling of Li^+ and OTf^- in the PHE. During the charging process, the Zn^{2+} in the mixture layer also owns the trend to migrate back to the APL side, though the sluggish kinetics of Zn^{2+} migration in such layer reduces the efficiency of Zn^{2+} diffusion, which at least promises the relative stability of the mixture layer with a dynamic Zn^{2+} ion concentration equilibrium.

We have followed your comment to add more discussions in the corresponding positions; please see highlighted parts on pages 9, 10 and 11 in the revised manuscript and page 11 in the supporting information.

Figure R3. a) Simulation of ions migration in the PHE (green sphere: Zn^{2+} , purple sphere: Li^+ , dark blue bond: OTf^- , light blue point: F atom from PVHF, red point: O atom from PMA); b) PMF of different ions moving from APL to CPL side.

Figure R4. Schematic illustration of ions migration mechanism in the PHE.

Comment 3: I do not quite understand Figure 3h and i, to be convincing authors should provide stabilities at much higher capacities.

Reply:

Thank you for the valuable comment. In our PHE electrolyte, the Zn^{2+} ions are effectively trapped in the APL sides. Thus, the stable Zn plating/stripping at the APL side and Li plating/stripping at CPL side can be achieved theoretically. Then, experimentally, we assembled the Li||Zn cell with the PHE to verify the design strategy (the theoretical output voltage of the Li||Zn cell should be 2.38 V, Li^+/Li^0 is -3.04 V vs. SHE and Zn^{2+}/Zn^0 is -0.76 V vs. SHE). First, as shown in the Figure R5, we explored the Zn and Li plating/stripping performance at different current densities, and stable Zn/Li deposition/dissolution with low polarization voltage can be maintained even at 2 mA cm^{-2} and 2 mAh cm^{-2} . Then, we followed your comment to conduct the long-term cycling of Li||Zn cells at high current density and high capacity (1 mA cm^{-2} and 1 mAh cm^{-2}) (Figure R6), in which stable 300 h cycling can be achieved at such high current density and capacity, furthering confirming the advance of the PHE design.

Following your comment, we have added more detailed discussions. Please see

highlighted parts on pages 10 and 11 in the revised manuscript.

Figure R5. GCD curves of Li||Zn cell at different current density based on the PHE.

Figure R6. Cycling performance of Li||Zn cell at 1 mA cm⁻² based on the PHE

Comment 4: literature typically don't discharge LNMO to low voltages so that to avoid Mn redox, but in Figure 4b the Mn redox is unclear (it should have roughly equivalent capacity as the Ni redox).

Reply:

Thank you for the valuable comment. Theoretically, during the discharge process of the LNMO electrode, the Mn redox is possible to be activated. However, the redox of Ni ($\text{Ni}^{4+}/\text{Ni}^{3+}$ and $\text{Ni}^{3+}/\text{Ni}^{2+}$) occurs before Mn redox due to the higher redox potential of Ni redox. Once the formation of $\text{LiNi}_{0.5}\text{Mn}_{1.5}\text{O}_4$ with the complete Ni redox, it is hard for Mn to be further reduced because there is no free site for Li^+ to be inserted into the spinel structure of LNMO. According to some previous reports (*J. Power Sources* **2012**, *215*, 312-316; *Proc. Natl. Acad. Sci.* **2018**, *115*, 1156-1161; *Nat. Commun.* **2022**, *13*, 1565.), even in the wide charge/discharge window (3.3-4.9 V, Li||LNMO cell), the Mn redox is still slight and the contributed capacity from Mn redox mainly can be attributed to the exposed Mn atom at the edge of the polycrystal LNMO. The inner Mn

atom in the crystal LNMO is hard to reduce due to the limited Li^+ sites, and such a phenomenon can also be detected in the similar LiCoMnO_4 electrode (*Chem* **2019**, *5*, 896-912.). For better clarity, we have added more discussion in the corresponding position. Please see the highlighted part on page 12 in the revised manuscript.

Comment 5: the stability of PHE and dynamics of the interface are essentially missing, and as such, it is unclear if the functional mechanisms proposed by the authors are valid.

Reply:

Thank you for the valuable comment. For the stability of the PHE, we first conducted the linear sweep voltammetry (LSV) test of the electrolyte (Figure R7). The PHE exhibits high electrochemical stability and is stable up to 2.9 V vs. Zn/Zn^{2+} , showing great promise for constructing a stable Zn||LNMO battery. Then, we verified the long-term stability of the PHE in Li|Zn cell with 1 mA cm^{-2} current density and 1 mAh cm^{-2} capacity (Figure R6), in which stable 300 h cycling for Zn/Li deposition/dissolution can be achieved, furthering confirming the stability of the PHE. Furthermore, we assembled the Zn||LNMO coin cell and pouch cell to clarify the improvement of cycling performance based on the PHE (Figure 4d and 5c in the revised manuscript), and benefiting from the remarkable stability of the PHE, the notable cycling performance of Zn||LNMO coin/pouch cells can be achieved. The well-maintained separated $\text{Zn}^{2+}/\text{Li}^+$ structure of the PHE after cycling can further confirm the stability of the PHE (inset picture in Figure 4d in the revised manuscript).

In addition, for the interface dynamics of the PHE, we first tested the PHE's resistance (Figure R8), and no clear interface charge transfer resistance can be detected, indicating the fast interface transfer of ions based on the PHE. Then, as shown in Figure R5, even after increasing the current density of the Li|Zn cell, the battery still can work properly with low polarization voltage. The fast interface transfer of ions in the PHE can also be confirmed by the excellent rate performance of the as-fabricated Zn||LNMO cell (Figure 4b and 4c in the revised manuscript).

We have followed your comment to add more data and discussions in the corresponding positions. Please see the highlighted parts on pages 10, 11 and 14 in the

revised manuscript and pages 11 and 12 in the supporting information.

Figure R7. Linear sweep voltammetry (LSV) curves of the Zn or Li|Ti cell based on the as-prepared SPEs (scan rate 0.5 mV s^{-1}).

Figure R8. EIS curves of ionic conductivity test of a) APL; b) CPL and c) PHE.

To Reviewer #2:

This manuscript reported polymer hetero-electrolytes for solid Zn batteries, which successfully block the Zn^{2+} shuttle and allow the separated Zn and Li reversibility. Benefiting from the unique polymer electrolyte, high-voltage lithium cathode is able to work in Zn/Li hybrid batteries. The as-resulted solid Zn/Li hybrid batteries can deliver significantly high output voltage compared with conventional zinc ion batteries, accompanying with the long cycling life and the great potential for large-scale application. The work is inspirable for constructing multi-ion hybrid system with tailored ion diffusion, and the strategy is potentially applicable for other metal ion batteries for achieving high-voltage and high-energy output. Thus, I recommend

publishing this work in Nature Communications. Some comments are as followed:

Reply:

Thank you for the positive comments on our work.

Comment 1: As solid electrolytes, the interface between the electrode and solid electrolytes cannot be ignored and it has a huge influence on the electrochemistry performance. Normally, some special treatments are necessary. How do the authors overcome this problem? Detailed procedures should be added in the experimental section.

Reply:

Thank you for the valuable comment. Indeed, the interface problem between electrodes and solid electrolytes severely limits the electrochemistry performance of solid batteries. In our work, to enhance the compatibility of electrodes and SPEs, we stored the as-fabricated solid batteries at 110 °C for 2h in order to weaken the interface problem. The supporting information has added the related experimental procedures on page 4.

Comment 2: High-voltage-type lithium cathode LNMO was employed in the hybrid battery and the battery shows remarkable cycling performance based on the polymer electrolytes. How about the high-voltage tolerance of the polymer electrolyte. What are the key factors affecting the high-voltage resistance of polymer electrolytes? It is due to the enhanced molecular interaction or the suppressed polymer chains kinetics in CPL?

Reply:

Thank you for the valuable comment. We have adopted linear sweep voltammetry (LSV) to investigate the anodic stability of the PHE (Figure S14 in the supporting information). The PHE exhibits high electrochemical stability and is stable up to 2.9 V vs. Zn/Zn²⁺. The key to achieving the remarkable high-voltage resistance of polymer electrolytes is improving the anodic stability of the polymer matrix. In our PHE, we crosslinked the polymethyl acrylate matrix with methacryl polyhedral oligomeric silsesquioxane (POSS) to effectively improve the structural stability due to the

enhanced molecular interaction. Thus, the resulting PHE can exhibit high anodic stability during cycling.

Comment 3: The Zn-LNMO cell was fabricated based on the as-prepared polymer hetero-electrolyte. The CE gradually got stable at above 99.6 % during the cycling. The CE should be compared with other Zn hybrid batteries reports to further reveal the decent ionic migration performance in the polymer electrolytes.

Reply:

Thank you for the valuable comment. We have followed your comment to compare the Coulombic efficiency (CE) of our Zn/LNMO battery with other Zn hybrid batteries, which was shown as Table R3. Benefiting from the fast ion migration kinetics and excellent stability in the PHE, the as-fabricated Zn/LNMO battery exhibits superior cycling performance with high CE compared with other batteries. We have followed your comment to add more data and discussion in the corresponding positions. Please see the highlighted parts on page 12 in the revised manuscript and page 16 in the supporting information.

Table R3. Comparison of the Zn hybrid batteries in different systems.

Cathodes	Electrolyte ^a	Capacity (mAh g ⁻¹)@Rate	Energy Density (Wh Kg ⁻¹)	Coulombic Efficiency (%)	Cycling life ^b	Ref
MgMn ₂ O ₄	1M MgSO ₄ + 1M ZnSO ₄ 0.5M	125(0.1 A g ⁻¹)	150	99.0	80% (500)@0.5 A g ⁻¹	2
LiV ₃ O ₈	LiOTf + 3M Zn(OTf) ₂ 1M	380(0.1 A g ⁻¹)	285	88.8	87% (4000)@5 A g ⁻¹	3
LiCoO ₂	Zn(OAc) ₂ + 4M LiOAc + NH ₃ H ₂ O	91(0.5 C)	173	98.7	95% (300)@2 C	4
FeHCF	3M KOTf + 3M	150(0.5 A g ⁻¹)	165	94.0	57% (1000)@1	5

	Zn(OTf) ₂ 21 M				A g ⁻¹	
LiVPO ₄ F	LiTFSI + 2M	125(0.1 A g ⁻¹)	237	95.0	87% (600)@1 A g ⁻¹	6
LiFePO ₄	Zn(OTf) ₂ 1M LiOTf + 1M	158(0.5 C)	174	94.0	89% (100)@1 C	7
Na ₃ V ₂ (PO ₄) ₃	Zn(OTf) ₂ 1M Ca(OTf) ₂ + 1M	81(1 C)	105	99.0	74% (1300)@2 0 C	8
LiMn ₂ O ₄	Zn(OTf) ₂ 1m Zn(TFSI) + 20m LiTFSI	65(0.2 C)	112	99.9	85% (4000)@4 C	9
Ag	0.1M ZnCl ₂	104(1.5 A g ⁻¹)	99	99.0	93% (1300)@1 A g ⁻¹	10
Graphite	3m Zn(TFSI) ₂ in ethyl methyl carbonate	110(0.1 A g ⁻¹)	231	94.0	96% (50)@0.1 A g ⁻¹	11
Bi ₂ O ₃	6M KOH + 0.3M Zn(OAc) ₂	323(0.3 C)	245	98.2	49% (1000)@1 0 A g ⁻¹	12
LNi _{0.5} Mn _{1.5} O ₄	PHE	113(0.05 C)	289	99.6	77.3% (450)@0.5 C	This Work

a. All the electrolytes are aqueous except special illustration.

b. The percentage means capacity retention, and the number in brackets means cycle numbers.

Comment 4: As shown in the Figure 4d and 5c, the cycling performance of LNMO exhibits obvious capacity increase at the beginning of cycling, what is the reason? For the high-voltage LNMO cathodes, what factors will result in the capacity decay during cycling, and whether the as-prepared polymer electrolytes can effectively suppress these issues?

Reply:

Thank you for the valuable comment. With the cycling proceeding, the compatibility between the electrode and polymer electrolyte will get improved. Thus the capacity of a solid battery normally exhibits a slight increase at the beginning of cycling (*Angew. Chem. Int. Ed.* **2020**, *132*, 11882-11886; *Adv. Mater.* **2022**, *34*, 2108665.). In addition, for the high-voltage LNMO electrode, the inevitable dissolution of transition metal cations during cycling will lead to severe capacity decay (*Nat. Commun.* **2022**, *13*, 1565.). Solid polymer electrolytes can effectively suppress the dissolution of transition metal cations, which can be confirmed by the as-fabricated Zn||LNMO battery with remarkable cycling stability (77.3% capacity retention after 450 cycles with high CE).

Comment 5: As the authors stated in the manuscript, it seems that there are multiple strategies to block Zn²⁺ motion in cathode side, including the selection of different polymer matrix in CPL and APL, the different thickness of CPL and APL, and the ion pairs effect of Zn²⁺ and Li⁺. What is the most important factor for achieving the highly blocked Zn²⁺ mobility, more explanations should be added in the manuscript.

Reply:

Thank you for the valuable comment. The key to achieving the highly trapped Zn²⁺ in the PHE is adopting the polymethyl acrylate (PMA) as the matrix in the CPL side. Due to the strong coordination between Zn²⁺ and carbonyl group in PMA, the Zn²⁺ can be effectively trapped at the interface between APL and CPL. To clearly clarify the influence of PMA on blocking Zn²⁺ on the CPL side, we conducted a molecular dynamic simulation to investigate the ion migration across the polymer-polymer interface between APL and CPL (Figure R9a). Li⁺ and OTF⁻ can freely shuttle through the polymer-polymer interface, but Zn²⁺ is strictly trapped at the interface between APL and CPL. The energy barrier of the Zn²⁺ ion migrating across the interface (52.8 kcal mol⁻¹) is much larger than Li⁺ (10.4 kcal mol⁻¹) and OTF⁻ (13.2 kcal mol⁻¹), which can confirm the efficiency of blocking Zn²⁺ with PMA matrix (Figure R9b). We have followed your comment to add more discussion in the corresponding positions. Please see the highlighted parts on pages 9, 10 and 11 in the revised manuscript.

Figure R9. a) Simulation of ions migration in the PHE (green sphere: Zn^{2+} , purple sphere: Li^+ , dark blue bond: OTf, light blue point: F atom from PVHF, red point: O atom from PMA); b) PMF of different ions moving from APL to CPL side.

Comment 6: In Figure 3j, the interface impedance of the Zn-Li battery gradually decreases, then further increases with the proceeding of cycling. It seems some side reactions happened. Please clarify the phenomenon.

Reply:

Thank you for the valuable comment. With the cycling proceeding, the interface resistance (R_i) of Li|Zn cell gradually decreases, which can be attributed to the improved interface compatibility between electrode and electrolyte. Subsequently, the battery exhibited a slight increase of the R_i , which can be attributed to the possible passivation on the Li metal electrode (*ACS Energy Lett.* **2019**, *4*, 690-701; *Angew. Chem. Int. Ed.* **2022**, *61*, e202209169.). Because in the following Zn||LNMO cell, there is no Li metal electrode incorporated and the as-fabricated battery exhibits remarkable cycling performance with stable Zn metal anode. We have followed your comment to add the discussion in the corresponding position. Please see the highlighted parts on page 11 in the revised manuscript and page 13 in the supporting information.

To Reviewer #3:

This work reports on fabricating hybrid, dual-layer, polymer electrolytes for Zn-Li batteries. The anode polymer layer (APL) consists of poly(vinylidene fluoride-co-hexafluoropropylene) (PVHF), Succinonitrile (SN) and Zn salt. In the cathode polymer layer, methacryl POSS/methyl acrylate was added to PVHF/SN/LiOTf. The motivation is to solve the problem of Zn²⁺ immobilization in LNMO cathodes. The design rationale is that the cathode layer blocks Zn²⁺ shuttling due to the strong coordination between PMA and Zn²⁺. Full cell tests were conducted to demonstrate the success of the design. Dual-layer SPE has been widely used in various secondary battery designs, and this one is another interesting application. This reviewer has the following questions about the electrolytes and battery design.

Reply:

Thank you for the positive comments on our work.

Comment 1: The goal is to solve the problem of immobilized Zn²⁺ in LNMO. However, when introducing crosslinked PMA, the Zn²⁺ ions are strongly coordinated with the polymer. This is basically equivalent to cation immobilization. Are Zn²⁺ ions still mobile in the battery? Judging from the extremely low conductivity and transference number, they seem immobile. If so, do we lose a significant amount of Zn into the electrolytes in every cycle? This could be a critical problem for the design. In the experiment, how thick was Zn foil? Would it work for thin or even anode-free design?

Reply:

Thank you for the insightful comment. We employed crosslinked PMA to inhibit the migration of Zn²⁺ into the cathode side. Thus, Zn²⁺ can only move in the APL side and cannot freely shuttle in the whole battery. By contrast, Li⁺ and OTf⁻ can freely migrate in the battery, and Li⁺ is the efficient charge carrier and OTf⁻ serves as the balance ion. To clearly reveal the ion migration at the polymer-polymer interface, we conducted the molecular dynamic (MD) simulation (Figure R10a). The simulation confirms that Li⁺ and OTf⁻ can freely shuttle through the polymer-polymer interface,

but Zn^{2+} is strictly trapped at the interface between APL and CPL. In addition, the potential of mean force (PMF) of Zn^{2+} moving across the interface vastly increases compared with Li^+ and OTf^- and the PMF of Zn^{2+} moving in CPL side is up to 52.8 kcal mol⁻¹ (Figure R10b).

Based on these observations, we proposed the possible mechanism of our PHE (Figure R11). When the Zn||LNMO battery is discharged, Zn^{2+} will start to move in the CPL direction, but due to the strong coordination between Zn^{2+} and carbonyl group in PMA, a mixture layer will form on the polymer-polymer interface, which mainly consists of the Zn salt with polymethyl acrylate from CPL side. Such a mixture layer inhibits the further diffusion of Zn^{2+} from the APL side and allows the free shuttling of Li^+ and OTf^- anion in the PHE, which can be confirmed by the low interface resistance based on the Bode plots of the batteries (Figure 3c in the revised manuscript and Figure S12 in the supporting information). In the subsequent charging process, the Zn^{2+} in the mixture layer also owns the trend to migrate back to the APL side, though the sluggish kinetics of Zn^{2+} migration in such layer reduces the efficiency of Zn^{2+} diffusion, which at least promises the relative stability of the mixture layer with a dynamic Zn^{2+} ion concentration equilibrium. Such a mechanism also can be confirmed by the SEM image of the PHE after cycling (inset picture in Figure 4d in the revised manuscript), in which well-trapped Zn^{2+} on APL side can be detected without diffusion into the cathode side. Meanwhile, the battery can work properly with 450 cycles of life-span.

In addition, in our battery, the thickness of the Zn anode is only 20 μm , which is much lower than conventional Zn hybrid batteries (above 100 μm) (*Energy Environ. Sci.* **2019**, *12*, 3288-3304.) (Table R4). It can be attributed to the stable solid polymer electrolyte that suppresses the dendrites and side reactions in the conventional Zn anode, thus effectively improving the Coulombic efficiency of the Zn anode in our battery (Figure R12). For anode-free design, it is difficult to be achieved in solid batteries due to the slow ion migration compared with liquid batteries, and such anode-free design need to overcome the stability issues during cycling. We may investigate the possibility of anode-free design in our system in the future.

Following your comment, we have added more detailed discussions. Please see

highlighted parts on pages 9, 10, 11 and 14 in the revised manuscript and page 11 in the supporting information.

Figure R10. a) Simulation of ions migration in the PHE (green sphere: Zn²⁺, purple sphere: Li⁺, dark blue bond: OTf, light blue point: F atom from PVHF, red point: O atom from PMA); b) PMF of different ions moving from APL to CPL side.

Figure R11. Schematic illustration of ions migration mechanism in the PHE.

Table R4. Comparison of our Zn||LNMO battery with the typical Zn hybrid batteries.

System	Electrolyte	The thickness of Zn anode (μm)	Separator	Output voltage (V)	Energy Density _a (node+cathode+electrolyte) (Wh L^{-1})	Ref
Zn LiMnO ₂ or LiFePO ₄	Liquid electrolyte with Zn + Li salts	>100	Glass fiber (>300 μm)	< 1.7	< 100	¹
Zn LNMO	PHE (~49 μm)	20	/	2.4	142	This work

Figure R12. GCD curves of Li|Cu cell for the CE test of metal Zn electrode.

Comment 2: Along the same line, does the Zn²⁺ concentration in PMA increase with cycling? This also affects lithium concentration, conductivity etc.

Reply:

Thank you for the valuable comment. Firstly, experimentally, we tested the SEM image of the PHE after cycling, which was shown as Figure R13. After the long-term cycling, the Zn²⁺ ions still can be well trapped in the APL side, indicating the superiority

of the design of PHE. Then, based on the MD simulation, we proposed the possible mechanism of the PHE, in which a mixture layer with Zn^{2+} incorporation at the polymer-polymer interface will form during the charging/discharging process. The Zn^{2+} ions trapped in the mixture layer on trend to move back to the APL side with a dynamic Zn^{2+} ion concentration equilibrium (Figure R10 and R11), assuring the stability of the PHE during cycling. We have added the discussions in the corresponding position. Please see the revised manuscript's highlighted parts on pages 10, 11, and 14.

Figure R13. SEM image and element mappings of the PHE after cycling (green dots: Zn^{2+} , red dots: Li^+).

Comment 3: How thick was each layer? At 120C, the film has a ~5% weight loss, which is considered the thermal stability limit. However, the sample processing was conducted at 150C. Based on TGA, there is ~10-20% weight loss already for the samples, which is way beyond the thermostability of the materials.

Reply:

Thank you for the valuable comment. The average thickness of APL and CPL is ~15 μm and ~35 μm , respectively. As observed from the TGA curves of our samples (Figure S2 in the supporting information), the weight loss before 120 °C is ~5.5% and the weight loss before 150 °C is ~8.8%. Considering that we prepared the test samples for TGA test in the ambient environment, the weight loss before 120 °C is mainly due to the residual water. Furthermore, considering the high decomposition temperature of PVHF (330 °C), PMA (320 °C), succinonitrile (190 °C), and POSS (460 °C) in the

polymer electrolyte, the weight loss before 150 °C may be attributed to the evaporation of the residual solvent DMF rather than the decomposition of components in PHE (*J. Power Sources* **2022**, 527, 231165.). To some extent, DMF is even beneficial for improving the ionic conductivity of polymer electrolytes. We have added more discussion in the corresponding position. Please see the highlighted parts on page 6 in the revised manuscript.

Comment 4: In Figure 4e, detailed conditions such as rates, temperature etc. should be given for a fair comparison.

Reply:

Thank you for the valuable comment. All the data referred to in the new Figure 4e in the revised manuscript (as shown below Figure R14) were collected at room temperature; we have followed your comment to mark this in the corresponding caption. In addition, we further summarized the performance, including the capacity at a detailed rate, the cycling life with detailed cycles and rate, and the corresponding electrolytes of these Zn hybrid batteries for a fair comparison, which was shown as Table R5. Our Zn||LNMO battery is superior to other Zn hybrid batteries, benefiting from the as-designed PHE. Please see highlighted parts on pages 12 and 14 in the revised manuscript and page 16 in the supporting information for the modification.

Figure R14. Comparison of the discharge voltage and capacity of Zn ion batteries and Zn hybrid batteries at room temperature (PANi: polyaniline; ZnHCF: zinc

hexacyanoferrate; C4Q: calix[4]quinone; NaFeHCF: sodium-iron hexacyanoferrate; FeHCF: iron hexacyanoferrate.).

Table R5. Comparison of the Zn hybrid batteries in different systems.

Cathodes	Electrolyte ^a	Capacity (mAh g ⁻¹)@Rate	Energy Density (Wh Kg ⁻¹)	Coulombic Efficiency (%)	Cycling life ^b	Ref
MgMn ₂ O ₄	1M MgSO ₄ + 1M ZnSO ₄ 0.5M	125(0.1 A g ⁻¹)	150	99.0	80% (500)@0.5 A g ⁻¹	2
LiV ₃ O ₈	LiOTf + 3M Zn(OTf) ₂ 1M	380(0.1 A g ⁻¹)	285	88.8	87% (4000)@5 A g ⁻¹	3
LiCoO ₂	Zn(OAc) ₂ + 4M LiOAc + NH ₃ H ₂ O 3M KOTf + 3M	91(0.5 C)	173	98.7	95% (300)@2 C	4
FeHCF	Zn(OTf) ₂ 21 M	150(0.5 A g ⁻¹)	165	94.0	57% (1000)@1 A g ⁻¹	5
LiVPO ₄ F	LiTFSI + 2M Zn(OTf) ₂ 1M LiOTf	125(0.1 A g ⁻¹)	237	95.0	87% (600)@1 A g ⁻¹	6
LiFePO ₄	+ 1M Zn(OTf) ₂ 1M	158(0.5 C)	174	94.0	89% (100)@1 C	7
Na ₃ V ₂ (PO ₄) ₃	Ca(OTf) ₂ + 1M Zn(OTf) ₂ 1m	81(1 C)	105	99.0	74% (1300)@20 C	8
LiMn ₂ O ₄	Zn(TFSI) + 20m LiTFSI	65(0.2 C)	112	99.9	85% (4000)@4 C	9
Ag	0.1M ZnCl ₂	104(1.5 A g ⁻¹)	99	99.0	93% (1300)@1 A g ⁻¹	10
Graphite	3m Zn(TFSI) ₂ in ethyl	110(0.1 A g ⁻¹)	231	94.0	96% (50)@0.1 A g ⁻¹	11

Bi_2O_3	methyl carbonate 6M KOH + 0.3M $\text{Zn}(\text{OAc})_2$	323(0.3 C)	245	98.2	49% (1000)@1 0 A g ⁻¹	12
$\text{LNi}_{0.5}\text{Mn}_{1.5}\text{O}_4$	PHE	113(0.05 C)	289	99.6	77.3% (450)@0.5 C	This Work

- a. All the electrolytes are aqueous except special illustration.
b. The percentage means capacity retention, and the number in brackets means cycle numbers.

To Reviewer #4:

In this paper, the authors deal with the polymer electrolyte design for Zn/Li hybrid batteries. The area of Zn battery as an emerging technology seeks ample attention. The most important aspect that the authors want to conclude in this study is to demonstrate that the polymer hetero-electrolyte (PHE), consisting of a anode layer for Zn^{2+} ion carries and a cathode layer for blocking Zn^{2+} ion carries, enables the sufficient cycling of 2.4 V $\text{LiNi}_{0.5}\text{Mn}_{1.5}\text{O}_4/\text{Zn}$ batteries. 40 mAh pouch cells are also prepared to convince the scale-up potential and low self-discharge property. Even though the research direction and the research problem are remarkably imperative, quite some flaws need to be addressed before the consideration of the manuscript for publication in this outstanding journal. Some important remarks are:

Reply:

Thank you for the comments on our work. We have followed your comments to revise our manuscript in the hope of addressing your concerns and meeting the high standard of this journal; please see below.

Comment 1: Does any other paper report about $\text{LiNi}_{0.5}\text{Mn}_{1.5}\text{O}_4/\text{Zn}$ batteries? As well as the polymer electrolyte for Zn/Li hybrid batteries? What is the most distinguish achievement here?

Reply:

Thank you for the valuable comment. There is no report about the Zn||LNMO battery upon our submission because it is very difficult to achieve effective Li⁺ intercalation/deintercalation in LNMO with Zn/Li hybrid electrolytes (new Figure 1e in the revised manuscript).

In addition, around seven papers have reported the polymer electrolytes with mixed bi-salts (Zn salt and Li salt) for Zn/Li hybrid batteries so far, and the employed cathodes in these batteries mainly are LiMn₂O₄ and LiFePO₄, leading to the low output voltage and energy density in such Zn/Li hybrid batteries (Figure R15). Other high-voltage cathodes including LNMO, NCM811 and LiCoO₂ are not able to be applied in the hybrid ion polymer electrolyte due to the irreversible Zn²⁺ insertion into electrodes (*Energy Storage Mater.* **2022**, *53*, 532-543; *ACS Appl. Energy Mater.* **2020**, *3*, 2526-2536.).

Our achievement here is achieving the high-voltage and rechargeable solid Zn||LNMO battery for the first time, which breaks the voltage and energy density limit of Zn/Li hybrid batteries with above 2.4 V output voltage and ~300 Wh Kg⁻¹ energy density. We have compared our battery with other Zn-based hybrid batteries (Figure R16 and Table R6), further confirming the superiority of our battery. The developed polymer hetero-electrolyte (PHE) here exhibits diverse advantages, including separated Zn²⁺/Li⁺ ions migration with fast kinetics based on the double-layer PHE, widened electrolyte window (anodic stability > 2.9 V vs. Zn²⁺/Zn), maintained mechanical strength at a low thickness (~49 μm) and high thermal stability. Furthermore, to verify the potential of our battery for practical application, the solid Zn||LNMO battery with high loading mass of LNMO (~8 mg cm⁻²) and thin Zn foil (20 μm) was fabricated with ~40 mAh output capacity, and the energy density of the pouch cell is ~142 Wh L⁻¹ calculated based on the anode, cathode and polymer electrolytes (the details of the calculations were shown as Table R7). These results indicate that the designed PHE paves a new way to develop high-voltage hybrid ion batteries based on reversible lithiated cathodes that conventionally cannot work properly in the mixed salts system.

We have added more discussions in the corresponding positions. Please see the highlighted parts on pages 3, 14, and 15 in the revised manuscript and pages 14, 16 and

17 in the supporting information.

Figure R15. Comparison of our work with Zn²⁺/Li⁺-separation polymer electrolyte and other Zn/Li hybrid batteries with hybrid ion polymer electrolytes (HCE-PAAm: highly concentrated dual-ion electrolyte- polyacrylamide; FS: fumed silica; PEG: polyethylene glycol; PVA: polyvinyl alcohol; PVP: polyvinyl pyrrolidone; GO: graphene oxide; SA: sodium alginate.)^{6, 13, 14, 15, 16, 17, 18}

Figure R16. Comparison of the discharge voltage and capacity of Zn ion batteries and Zn hybrid batteries at room temperature (PANi: polyaniline; ZnHCF: zinc hexacyanoferrate; C4Q: calix[4]quinone; NaFeHCF: sodium-iron hexacyanoferrate; FeHCF: iron hexacyanoferrate.).

Table R6. Comparison of the Zn hybrid batteries in different systems.

Cathodes	Electrolyte ^a	Capacity (mAh g ⁻¹)@Rate	Energy Density (Wh Kg ⁻¹)	Coulombic Efficiency (%)	Cycling life ^b	Ref
MgMn ₂ O ₄	1M MgSO ₄ + 1M ZnSO ₄ 0.5M	125(0.1 A g ⁻¹)	150	99.0	80% (500)@0.5 A g ⁻¹	2
LiV ₃ O ₈	LiOTf + 3M Zn(OTf) ₂ 1M	380(0.1 A g ⁻¹)	285	88.8	87% (4000)@5 A g ⁻¹	3
LiCoO ₂	Zn(OAc) ₂ + 4M LiOAc + NH ₃ H ₂ O	91(0.5 C)	173	98.7	95% (300)@2 C	4
FeHCF	3M KOTf + 3M Zn(OTf) ₂ 21 M	150(0.5 A g ⁻¹)	165	94.0	57% (1000)@1 A g ⁻¹	5
LiVPO ₄ F	LiTFSI + 2M Zn(OTf) ₂ 1M LiOTf	125(0.1 A g ⁻¹)	237	95.0	87% (600)@1 A g ⁻¹	6
LiFePO ₄	+ 1M Zn(OTf) ₂ 1M	158(0.5 C)	174	94.0	89% (100)@1 C	7
Na ₃ V ₂ (PO ₄) ₃	Ca(OTf) ₂ + 1M Zn(OTf) ₂ 1m	81(1 C)	105	99.0	74% (1300)@2 0 C	8
LiMn ₂ O ₄	Zn(TFSI) + 20m LiTFSI	65(0.2 C)	112	99.9	85% (4000)@4 C	9
Ag	0.1M ZnCl ₂ 3m	104(1.5 A g ⁻¹)	99	99.0	93% (1300)@1 A g ⁻¹	10
Graphite	Zn(TFSI) ₂ in ethyl methyl carbonate	110(0.1 A g ⁻¹)	231	94.0	96% (50)@0.1 A g ⁻¹	11
Bi ₂ O ₃	6M KOH + 0.3M	323(0.3 C)	245	98.2	49% (1000)@1	12

	Zn(OAc) ₂				0 A g ⁻¹	This Work
LNi _{0.5} Mn _{1.5} O ₄	PHE	113(0.05 C)	289	99.6	77.3% (450)@0.5 C	

- a. All the electrolytes are aqueous except special illustration.
b. The percentage means capacity retention, and the number in brackets means cycle numbers.

Table R7. The calculation process of energy density of the Zn||LNMO batteries.

V (Average voltage)	C_s (specific capacity)	m_A (loading mass of active materials)	V_T (volume of batteries based on the anode, cathode and PHE)
2.35 V	63 mAh g ⁻¹	0.65 g	0.68 cm ⁻³
$E_V = \frac{V * C_s * m_A}{V_T}, 142 \text{ Wh L}^{-1}$			

Comment 2: The authors have to clearly mention the precedents on the major milestones to probe out the cycling of LiNi_{0.5}Mn_{1.5}O₄/Zn batteries, pointing to the novelty of the present attempt.

In the introduction part, the authors have to explain the different electrolyte design of Li/Zn hybrid batteries, indicating the novelty of PHE beyond the previous research.

Reply:

Thank you for the valuable comment. Currently, no research is achieving the Zn||LNMO battery, and we developed the first highly reversible Zn||LNMO battery. We have followed your suggestion to mention this in the introduction part.

For the Zn/Li hybrid batteries, the common electrolytes can be divided into two parts: liquid electrolytes and hydrogel electrolytes. All the electrolytes are a mixture of Zn salts and Li salts in a solvent or polymer. The commonly used cathodes in Zn/Li hybrid batteries are low-voltage LiMn₂O₄ and LiFePO₄, which can work properly in the above electrolyte. The developed hydrogel electrolytes mainly aim at eliminating

the dendrites and side reactions at the Zn anode side. However, all the electrolytes for common Zn/Li hybrid batteries are unsuitable for constructing the Zn||LNMO battery because the shuttling and intercalation of Zn^{2+} in the LNMO electrode cannot be avoided. Then, we developed a unique PHE consisting of an anode layer with Zn^{2+} ions for $Zn^{0/2+}$ reactions and a cathode layer that can fully block the Zn^{2+} ion shuttle, leading to the highly reversible Zn||LNMO battery.

Following your comment, we added more description in the introduction to indicate the novelty of our designed electrolyte and battery. Please see the revised manuscript's highlighted parts on pages 3 and 4.

Comment 3: The authors mentioned that "Simple separation of Zn^{2+} and Li^+ ions by two polymer membranes is hard ..." Here, the authors didn't cite any literature or provide any specific explanation to reach the postulate.

Reply:

Thank you for the valuable comment. In Figure 1e in the revised manuscript, we directly attached one polymer membrane containing Zn salt to another polymer membrane containing Li salt, then employed the combined composite membranes as polymer electrolytes for constructing the Zn||LNMO battery. The resulting battery can only work for a few cycles with fast capacity decay, which obviously can be attributed to the shuttling of Zn^{2+} to the cathode side. Thus, we concluded that the simple separation of Zn^{2+} and Li^+ ions by two polymer membranes makes it hard to suppress the Zn^{2+} shuttling. We have added more explanation and the necessary reference in the corresponding position. Please see the highlighted part on page 4 in the revised manuscript.

Comment 4: What is the Coulombic efficiency of metallic Zn in the designed PHE?

Reply:

Thank you for the valuable comment. We have assembled a Li|Cu cell to investigate the Coulombic efficiency (CE) of metal Zn based on the as-designed PHE, which was shown as Figure R17. At the Li side, the Li plating/stripping occurs, and at

the Cu side, the Zn plating/stripping occurs, respectively, and such an electrochemical test can obtain the CE of metal Zn accurately. The LilCu cell based on the PHE works properly, and the CE of metal Zn is ~99.1%, indicating the high stability and efficiency of PHE. We have added more discussion in the corresponding position. Please see the highlighted parts on pages 10 and 11 in the revised manuscript.

Figure R17. GCD curves of LilCu cell for the CE test of metal Zn electrode.

Comment 5: Besides molecular dynamic simulation in Figure 3, is there other specific characterization to prove Zn²⁺ ions are more encased in the PMA chain than Li⁺ ions.

Reply:

Thank you for the insightful comment. First, we tested the ionic conductivity of Zn²⁺ and Li⁺ ions in different polymer matrices, including PMA and PVHF (Figure R18). The ionic conductivity of Zn²⁺ ions in PMA is $5.9 \times 10^{-7} \text{ S cm}^{-1}$, which is far below the ionic conductivity of Li⁺ ions in PMA ($3.5 \times 10^{-5} \text{ S cm}^{-1}$), confirming the sluggish migration kinetics of Zn²⁺ in PMA matrix. Then, as observed from the cross-sectional SEM image of PHE after cycling (Figure R19), the Zn²⁺ is still well trapped in the APL side, further suggesting the eliminated diffusion of Zn²⁺ in PMA-based CPL. We have added more discussion in the corresponding position. Please see the highlighted parts on page 7 in the revised manuscript.

Figure R18. Ionic conductivities (σ) of Zn^{2+} and Li^+ ions in different polymer matrix.

Figure R19. SEM image and element mappings of the PHE after cycling (green dots: Zn^{2+} , red dots: Li^+).

Comment 6: In figure 4d, the X axis should be "Specific capacity" rather than "Voltage".

Reply:

Thank you for the valuable comment. We are sorry for the mistake, and we have modified Figure 4d in the revised manuscript. Please check.

Comment 7: As mentioned by the authors, "... the energy density is projected to around 300 Wh kg⁻¹ ..." Here, is the energy density only based on active material in

anode/cathode side? A detailed calculation method should be provided in the SI. Besides Figure 4e, it is also better to provide a comparison figure for energy density differences in different Zn (hybrid) batteries.

Reply:

Thank you for the valuable comment. The 300 Wh Kg⁻¹ energy density described here is based on the active materials. For the practical application, we fabricated the pouch cell-level Zn||LNMO battery, shown in Figure 5c in the revised manuscript. The energy density of the pouch cell is ~142 Wh L⁻¹ calculated based on the anode, cathode, and polymer electrolytes, and the details of the calculations were shown as Table R8 in the supporting information.

In addition, we have followed your comment to add more details of the comparison between our battery and other Zn hybrid batteries, including energy density, rates, cycling life, and the adopted electrolytes system, which was shown as Table R9 in the supporting information. Please see the highlighted part on pages 12 and 13 in the revised manuscript and pages 16 and 17 in the supporting information.

Table R8. The calculation process of energy density of the Zn||LNMO batteries.

V (Average voltage)	C_s (specific capacity)	m_A (loading mass of active materials)	V_T (volume of batteries based on the anode, cathode and PHE)
2.35 V	63 mAh g ⁻¹	0.65 g	0.68 cm ⁻³
$E_V = \frac{V * C_s * m_A}{V_T}, 142 \text{ Wh L}^{-1}$			

Table R9. Comparison of the Zn hybrid batteries in different systems.

Cathodes	Electrolyte ^a	Capacity(mAh g ⁻¹)@Rate	Energy Density (Wh Kg ⁻¹)	Coulombic Efficiency (%)	Cycling life ^b	Ref
MgMn ₂ O ₄	1M MgSO ₄	125(0.1)	150	99.0	80%	²

	+ 1M ZnSO ₄ 0.5M	A g ⁻¹)			(500)@0.5 A g ⁻¹	
LiV ₃ O ₈	LiOTf + 3M Zn(OTf) ₂ 1M	380(0.1 A g ⁻¹)	285	88.8	87% (4000)@5 A g ⁻¹	3
LiCoO ₂	Zn(OAc) ₂ + 4M LiOAc + NH ₃ H ₂ O 3M KOTf + 3M	91(0.5 C)	173	98.7	95% (300)@2 C	4
FeHCF	Zn(OTf) ₂ 21 M	150(0.5 A g ⁻¹)	165	94.0	57% (1000)@1 A g ⁻¹	5
LiVPO ₄ F	LiTFSI + 2M Zn(OTf) ₂ 1M LiOTf	125(0.1 A g ⁻¹)	237	95.0	87% (600)@1 A g ⁻¹	6
LiFePO ₄	+ 1M Zn(OTf) ₂ 1M	158(0.5 C)	174	94.0	89% (100)@ 1 C	7
Na ₃ V ₂ (PO ₄) ₃	Ca(OTf) ₂ + 1M Zn(OTf) ₂ 1m	81(1 C)	105	99.0	74% (1300)@2 0 C	8
LiMn ₂ O ₄	Zn(TFSI) + 20m LiTFSI	65(0.2 C)	112	99.9	85% (4000)@4 C	9
Ag	0.1M ZnCl ₂ 3m Zn(TFSI) ₂ in ethyl methyl carbonate	104(1.5 A g ⁻¹)	99	99.0	93% (1300)@1 A g ⁻¹	10
Graphite	6M KOH + 0.3M Zn(OAc) ₂	110(0.1 A g ⁻¹)	231	94.0	96% (50)@0.1 A g ⁻¹	11
Bi ₂ O ₃		323(0.3 C)	245	98.2	49% (1000)@1 0 A g ⁻¹	12
LNi _{0.5} Mn _{1.5} O ₄	PHE	113(0.05 C)	289	99.6	77.3% (450)@0.5 C	Thi s Wo rk

a. All the electrolytes are aqueous except special illustration.

b. The percentage means capacity retention, and the number in brackets means cycle

numbers.

Comment 8: What is the main reasons for the initial capacity increase in both coin and pouch cells? (Figure 4d and 5c) Why is the capacity 10 mAh g⁻¹ less in pouch cells than coin cells under the same rate?

Reply:

Thank you for the valuable comment. In solid batteries, the compatibility between the electrode and polymer electrolyte will improve with the cycling proceeding. Thus, the capacity of a solid battery normally exhibits a slight increase at the beginning of cycling (*Angew. Chem. Int. Ed.* **2020**, *132*, 11882-11886; *Adv. Mater.* **2022**, *34*, 2108665.). In addition, due to the high loading mass of active materials (8 mg cm⁻²) in the pouch cell compared with the coin cell (1.5 mg cm⁻²), the ion diffusion kinetics in the LNMO thick electrode will slightly decrease, leading to the reduced utilization ratio of active materials (*Adv. Mater.* **2023**, *35*, 2209074). We have added a detailed explanation in the corresponding positions. Please see the highlighted parts on page 12 in the revised manuscript.

Comment 9: The overall style of presentation and quite some sentences in the manuscript has to be further modified.

Reply:

Thank you for the valuable comment. Following your comment, we have thoroughly checked our manuscript and modified our article to make it more concise and clearer. Please see the highlighted part on pages 3, 4, 6, 9 and 12 in the revised manuscript.

REVIEWER COMMENTS

Reviewer #1 (Remarks to the Author):

I appreciate authors consideration of my comments. Although I agree this manuscript is in better shape now, unfortunately I still can not see sufficient level of novel insights I would expect from papers in this journal and I still feel this design is somewhat fundamentally limited. The observation of ZnMn_2O_4 is intriguing and should be better supported, as this means the LNMO, usually considered as a stable spinel, somehow decomposed even with the presence of Li ions. The interfaces of two polymers are important and should be examined much more carefully. It is unclear how ions diffuses across the boundary layer, and if so, what is the maximum capacity that this layer can supports. In the cases of high capacity cycling, there must be some cross-diffusion but not much information is provided regarding how this layer looks like after long cycling. The discussions regarding high energy density is very misleading, and should be supported by realistic numbers that the authors actually experimentally use.

Reviewer #2 (Remarks to the Author):

The authors show a well-deserved response to my comments. At this state, I have no doubt in this work. The revised manuscript can be accepted in the present form.

Reviewer #3 (Remarks to the Author):

The revised manuscript can be accepted.

Reviewer #4 (Remarks to the Author):

The authors have addressed all the concerns mentioned by the reviewers, so this version of manuscript could be accepted now.

Response to Reviewers

Dear Reviewer,

Thanks a lot for your constructive comments and suggestions about our manuscript entitled “Polymer Hetero-Electrolyte Enabled Solid-state 2.4-V Zn/Li Hybrid Batteries” (NCOMMS-23-15899A), which we really appreciate. All comments are greatly valuable and helpful for improving the quality of our paper. We have studied your comments carefully and accordingly revised our manuscript in hope of addressing your concerns and meeting the high standards of *Nature Communications*. The comments were addressed point-by-point below and the related changes have been highlighted in the revised manuscript and supporting information.

To Reviewer #1:

I appreciate authors consideration of my comments. Although I agree this manuscript is in better shape now, unfortunately I still can not see sufficient level of novel insights I would expect from papers in this journal and I still feel this design is somewhat fundamentally limited.

Reply:

Thank you for the valuable comment. LNMO as a high-voltage cathode is promising for enhancing the energy output of Zn hybrid batteries. However, it is very difficult to achieve the stable and reversible cycling of LNMO in Zn hybrid batteries due to the irreversible intercalation of Zn^{2+} ions. Thus, our development reported in the manuscript achieved a high-voltage and rechargeable solid Zn||LNMO battery, which breaks the voltage and energy density limit of Zn/Li hybrid batteries with above 2.4 V output voltage. We designed the polymer-hetero electrolyte (PHE) with two polymer layers by employing the crosslinked poly (methyl acrylate) (PMA) to inhibit the migration of Zn^{2+} ions. The concept of inhibiting Zn^{2+} ions from reaching the cathode proved to be feasible, and the PHE electrolyte provided the prototype for using this

method, which can improve the energy output of Zn hybrid batteries. The concept developed can also be applied in other hybrid systems. We have provided a detailed reply to address your concerns well; please see the following response.

Comment 1: The observation of ZnMn₂O₄ is intriguing and should be better supported, as this means the LNMO, usually considered as a stable spinel, somehow decomposed even with the presence of Li ions.

Reply:

Thank you for the valuable comment. In the previous manuscript, we have provided the XRD pattern of the electrode after cycling (Figure 1c in the revised manuscript), which can serve as direct and strong evidence to confirm the existence of ZnMn₂O₄. Furthermore, we have followed your comment to provide more evidence to confirm the existence of ZnMnO₂. First, we collected the X-ray photoelectron spectroscopy (XPS) of the electrodes at the initial state and after cycling, which were shown in Figure R1. An obvious signal of Zn 2p can be detected in the electrode after cycling, indicating the intercalation of Zn²⁺ ions during cycling. In addition, compared with the electrode at initial state, the increased content of low-valence Mn (ZnMn₂O₄) and high-valence Ni (LiNiO₂) in the electrode after cycling can further confirm the irreversible intercalation of Zn²⁺ ions, which agrees well with the above XRD result. Then, we also collected the high-resolution transmission electron microscopy (HRTEM) images of the electrode after cycling, which is shown in Figure R2. The obvious crystalline graph of ZnMn₂O₄ can be detected on the surface of the LNMO particle, indicating the irreversible transition of the LNMO cathode after the intercalation of Zn²⁺ ions. Thus, the existence of ZnMn₂O₄ can be fully confirmed by the above evidence. We have followed your comment to add more discussions in the corresponding positions; please see highlighted parts on page 4 in the revised manuscript and pages 5 and 6 in the supporting information.

Figure R1. XPS spectra of the electrodes at the initial state and after cycling: a) Zn 2p, b) Mn 2p and c) Ni 2p.

Figure R2. a) and b) HRTEM images with different magnifications of the LNMO electrode after cycling.

Comment 2: The interfaces of two polymers are important and should be examined much more carefully. It is unclear how ions diffuse across the boundary layer, and if so, what is the maximum capacity that this layer can support. In the cases of high capacity cycling, there must be some cross-diffusion but not much information is provided regarding how this layer looks like after long cycling.

Reply:

Thank you for the insightful comment. In the design of the polymer-hetero electrolyte (PHE), we employed crosslinked poly (methyl acrylate) (PMA) to inhibit the migration of Zn^{2+} ions into the cathode side. Thus, Zn^{2+} ions can only move in the anode polymer layer (APL) side and cannot freely shuttle in the whole battery. By contrast, Li^+ cations and OTf^- anions can freely migrate in the battery, and Li^+ ions are

the efficient charge carriers, and OTf⁻ ions serve as the balance ion. To reveal the ion migration at the polymer-polymer interface, we conducted the molecular dynamic (MD) simulation (Figure 3a-e in the revised manuscript). The simulation confirms that Li⁺ ions and OTf⁻ ions can freely shuttle through the polymer-polymer interface, but Zn²⁺ ions are strictly trapped at the interface between APL and cathode polymer layer (CPL).

Then, we proposed the possible mechanism of our PHE (Figure 3f in the revised manuscript). When the Zn||LNMO battery is discharged, Zn²⁺ ions will start to move in the CPL direction, but due to the strong coordination between Zn²⁺ ions and carbonyl group in PMA, a mixture layer will form on the polymer-polymer interface, which mainly consists of the Zn salt with PMA from CPL side. Such a mixture layer inhibits the further diffusion of Zn²⁺ from the APL side and allows the free shuttling of Li⁺ ions and OTf⁻ anions in the PHE. In the subsequent charging process, the Zn²⁺ ions in the mixture layer also own the trend to migrate back to the APL side, though the sluggish kinetics of Zn²⁺ ions migration in such layer reduces the efficiency of Zn²⁺ diffusion, which at least promises the relative stability of the mixture layer with a dynamic Zn²⁺ ions concentration equilibrium.

Thus, the migration and the efficiency of plating/stripping of Zn²⁺ ions are key and directly affect the limit of the output capacity of the battery with PHE. We fabricated the Li||Cu cell to investigate the plating/stripping performance of Zn²⁺ ions with different capacities, which was shown in Figure R3. Stable plating/stripping of Zn²⁺ ions can be maintained when the capacity is below 3 mAh cm⁻². However, when the capacity increased to 4 mAh cm⁻², unstable plating/stripping of Zn²⁺ ions can be detected with low Coulombic efficiency (CE), which can be attributed to the increased diffusion impedance of Li⁺ ions and OTf⁻ ions due to the mixture layer at the polymer interface. The CE of the plating/stripping of Zn²⁺ ions is 99.3% (1 mAh cm⁻²), 98.7% (2 mAh cm⁻²), 95.1% (3 mAh cm⁻²) and 62.2% (4 mAh cm⁻²), respectively. Thus, the maximum capacity that our PHE can support is below 3 mAh cm⁻². It is worth noting that the current cycling capacity of the solid batteries is normally below 2 mAh cm⁻² (Nat. Energy 2023, 8, 230; J. Energy Storage 2023, 73, 109048.), therefore our developed PHE can sustain the proper operation of the common solid batteries.

In addition, we agree with your comment that the ion cross-diffusion exists during cycling. We provided the cross-sectional SEM image and elemental mappings of the PHE after cycling, shown in Figure R4. The double-layer structure of the PHE remained unchanged, indicating the decent stability of the solid electrolyte. The diffusion of Zn^{2+} ions at the interface of the two layers can be detected, resulting in the formation of a mixture layer area. The Zn^{2+} ions diffusion in the CPL side is not that severe, which can be attributed to the relatively areal capacity ($\sim 0.5 \text{ mAh cm}^{-2}$) of the battery. We have followed your comment to add more discussions in the corresponding positions; please see highlighted parts on page 11, 13 and 15 in the revised manuscript and page 14 in the supporting information.

Figure R3. a) The galvanostatic charge/discharge (GCD) curves of the Li|Cu cells at different capacities and b) the corresponding CE of the cells.

Figure R4. a) SEM image of the cross-sectional PHE after cycling and b) the corresponding elemental mapping image.

Comment 3: The discussions regarding high energy density is very misleading, and should be supported by realistic numbers that the authors actually experimentally use.

Reply:

Thank you for the valuable comment. We are sorry for the previous insufficient presentation of the discussion of energy density. To fully verify the potential of our battery for practical application, the solid Zn||LNMO battery with high loading mass of LNMO ($\sim 8 \text{ mg cm}^{-2}$) and thin Zn foil ($20 \text{ }\mu\text{m}$) was fabricated with $\sim 40 \text{ mAh}$ output capacity, and the calculated energy density of the pouch cell is $\sim 142 \text{ Wh L}^{-1}$ calculated based on the anode, cathode and polymer electrolytes (the details of the calculations were shown as Table R1). However, when considering the package of the pouch cell, the energy density is as low as 61 Wh L^{-1} . Following your comment, we have added more detailed discussions. Please see highlighted parts on page 16 in the revised manuscript and page 18 in the supporting information.

Table R1. The calculation process of energy density of the Zn||LNMO batteries.

$V_{\text{(Average voltage)}}$	$C_{\text{s(specific capacity)}}$	$m_{\text{A(loading mass of active materials)}}$	$V_{\text{T(volume of batteries based on the anode, cathode and PHE)}}$	$V_{\text{T(volume of the whole cell)}}$
2.35 V	63 mAh g ⁻¹	0.65 g	0.68 cm ⁻³	1.78 cm ⁻³
$E_V = \frac{V * C_s * m_A}{V_T}$, 142 Wh L ⁻¹ (based on the anode, cathode and PHE) and 54 Wh L ⁻¹ (based on the whole cell)				

REVIEWERS' COMMENTS

Reviewer #1 (Remarks to the Author):

I appreciate authors consideration of my comments, I don't have other comments.

Response to Reviewers

Dear Reviewer,

Thanks a lot for your constructive comments about our manuscript entitled “Polymer Hetero-Electrolyte Enabled Solid-state 2.4-V Zn/Li Hybrid Batteries” (NCOMMS-23-15899B), which we really appreciate. The comments were addressed point-by-point below.

To Reviewer #1:

I appreciate authors consideration of my comments, I don't have other comments.

Reply:

We deeply thank the reviewer's comment.